# Demographic and behavioural drivers of intra-urban mobility of migrant street children and youth in Kampala, Uganda

**Mulekya Francis Bwambale**[1,2]*, **Paul Bukuluki**[2], **Cheryl A. Moyer**[3], **Bart H. W. van den Borne**[1]

**1** Department of Health Education & Promotion, University of Maastricht Care and Public Health Research Institute (CAPHRI), Faculty of Health, Medicine and Life Sciences, Maastricht, Netherlands, **2** Department of Social Work and Social Administration, School of Social Sciences, Makerere University, College of Humanities and Social Sciences, Kampala, Uganda, **3** Departments of Learning Health Sciences and Obstetrics and Gynaecology, University of Michigan Medical School, Ann Arbor, Michigan, United States of America

* francisbmf@gmail.com

**Data Availability Statement:** The Minimal Dataset on which the findings and conclusions of the study are based has been included in the Supporting Information files. For ethical reasons, all variables

## Abstract

While research on the nexus of migration and wellbeing of individuals has gained recognition in recent years, far less attention has been devoted to intra-urban mobility especially among the urban poor young populations. We assess the drivers of intra-urban mobility using a random sample of 412 migrant street children and youth in Kampala city, Uganda. This paper draws from a larger cross-sectional survey of circular migration and sexual and reproductive health choices among street children in Kampala, Uganda. We define 'migrants' as street children and youth with a rural-urban migration experience and 'intra-urban mobility' as the number of places stayed in or moved since migrating to the city, measured on a continuous scale. More than half (54.37%) of the migrant street children and youth had lived in two or more places since migrating to the city. Multivariate negative binomial regression analysis reveals migrant street children and youth's intra-urban mobility to be associated with gender (aIRR = 0.71, 95%CI 0.53–0.96), sex work (aIRR = 1.38, 95%CI 1.01–1.88), a daily income of one USD or more (aIRR = 1.57, 95%CI 1.16–2.13) and duration of stay in the city (aIRR = 1.54, 95%CI 1.17–2.01). Other drivers of intra-urban mobility included availability of causal work, personal safety and affordability of rental costs. Our findings suggest the need for urban housing and health policies to take into account street children and youth's intra-urban mobility and its drivers. Future research on all drivers of street children and youth's intra-urban mobility and its linkage with their health outcomes is recommended.

## Introduction

Research on intra-urban mobility and well-being of individuals has gained attention over recent decades, with most available literature focused on developed countries [1, 2]. Intra-urban mobility or intra-urban migration, is often defined as the frequency with which

that may erode the privacy and confidentiality of the respondents have been dropped from the dataset.

**Funding:** We would like to acknowledge the Maastricht University, Care and Public Health Research Institute (CAPHRI) for co-funding this study and the Applied Research Bureau, Kampala, Uganda for supporting the training of research assistants.

**Competing interests:** The authors have declared that no competing interests exist.

individuals or families change their residence [3, 4]. The term is also generalised to short distance intra-community moves within city spaces [5]. At the individual level, intra-urban residential mobility has been re-conceptualized as relational practices that link lives through time, multi-locality and space while connecting people to structural conditions [6, 7]. Evidence shows that choice to change a location is influenced by the individuals' satisfaction or dissatisfaction with respect to utilities in a given location [8, 9]. For instance, migrants who rent public housing as their initial housing choice are much less likely to experience mobility compared to the long-term migrants who seem to gain some degree of residential stability [10]. Socioeconomic and individual statuses such as age, sex and tenure remain important antecedents of daily mobility [11].

Childhood intra-urban mobility has been shown to be related to poor self-reported health outcomes and well-being in adolescence and late adulthood [12]. Emerging research indicates that frequent changes of residence during childhood are associated with increased risk of numerous developmental and behavioural problems among children, including substance abuse, sex work and sexual promiscuity especially among the unstably housed persons [13]. Residential instability may influence sexual and drug-related behaviour and lack of power within survival sex especially among those living in uncertain circumstances [14].

Street children have been defined as any individuals for whom the street (including unoccupied dwellings) have become their place of living and/or source of livelihood, inadequately protected and supervised by responsible adults [15, 16]. While a child is legally defined as a person aged less than 18 years, the legal definition of a street child in Uganda does not exist [17]. Anecdotal evidence estimates between 10,000 and 20,000 street children in Uganda, with Kampala, the capital and largest city having the highest number of street children [18, 19]. In this study, we operationally define street children and youth as persons aged 12–24 years who spent most of their time on the streets and for whom the street is the main source of livelihood [20].

While research on the nexus of migration and wellbeing of individuals has gained recognition in recent years, far less attention has been devoted to intra-urban mobility of the urban poor especially street children and youth. Also, it is unclear how intra-urban mobility varies between short- and long-term migrant street children and youth. Understanding intra-urban mobility and its drivers among street children and youth with a rural-urban migration experience is critical in designing targeted interventions to improve their personal safety and social services within the city spaces. Street children and youth's safety in the urban spaces is critical for their health and wellbeing. Within the aspirations of the Uganda Vision 2040, the increasing trends in rural-urban and inter-city migration of the young people makes it imperative to ensure well planned urban/rural settlements and services and strong social safety nets for the vulnerable groups [21].

We examined data from a larger cross-sectional study which aimed to assess the nexus of circular migration and reproductive health choices among street children and youth in Kampala, Uganda. In this paper, we assess demographic and behavioural drivers of intra-urban mobility among a sub-set of migrant street children and youth who are of interest to our study. This study is positioned within the context of internal rural–urban migration that has resulted into dramatic influxes in the number of urban residents especially street children and youth in Ugandan cities in recent years [22]. The main interest of the study is on understanding intra-urban mobility of migrant and not lifelong native street children and youth. We theorize that lifelong native street children and youth are likely to have more stable settlements and support systems and therefore not worth studying their intra-urban mobility. While previous studies and models typify intra-urban residential mobility based on defined housing and neighborhood addresses [23], our study models the drivers of intra-urban mobility for street

children and youth. Within the local study context, we define intra-urban mobility in terms of the multiple places stayed in over time, using the question: "*Since you came to settle in the city, how many places have you stayed or lived in*?". The choice of intra-urban mobility as the outcome of interest is relevant to scholars of migration and health in cities as well as urban authorities, who grapple with the issue of intra-urban mobility of street children. We then use the negative binomial regression and Poisson regression and Ordinary Least Squares (OLS) as alternative models to explain the drivers of intra-urban mobility among migrant street children and youth, while controlling for confounding.

## Materials and methods

### Study setting

Data were collected using a survey conducted in three of five divisions of Kampala between April and July 2019. The three divisions of Makindye, Central and Rubaga were purposively selected based on the presence of stable congregation venues for street children and youth guided by the pre-survey mapping exercise. Kampala, the capital city of Uganda, serves as the major commercial centre in the country with approximately 1,583,000 residents, of which 28.2% are aged 15 to 24 years with an annual growth rate of 2% [24]. Kampala hosts the highest number of street children and serves as a major migration hub in Uganda, characterized by vibrant rental markets and widespread landlordism.

### Study population

While the total sample size of the study was 513, we limit our analysis to migrant street children and youth who constituted 80.31% (n = 412) of the total survey sample size. We enlisted street children and youth aged 12–24 years who; a) had stayed in the city at least 3 months prior to the survey, b) spent most of their time on the streets of Kampala and c) for whom assent and/or consent were obtained. Participants unable to provide the required information either due to illness, such as those experiencing obvious mental health problems, were excluded [25]. The choice of the age group 12–24 years as the study population was informed by the authors' understanding of the local context of street life in Kampala, in which some street children and youth have spent considerable time and transitioned to adults on the streets. Similar age categorisations have been used elsewhere [26].

### Sampling strategy

Street children and youth were selected using the venue-based time-space (VBTS) sampling technique. VBTS is recommended when all population members can be reached at certain locations at different times and where no comprehensive census data of the target population exists and has been used by earlier work [27–29]. First, to identify the parishes and geographic locations/venues where street children and youth congregate during daytime and could be contacted, we conducted a rapid mapping exercise in three of five purposively selected divisions of Kampala capital city. The parishes served as the primary sampling units for data collection and were selected using proportionate probability. That is, the size of the sample of street children and youth selected from each parish was proportional to the number of congregation venues situated in that parish. Notably, we found it illogical to include parishes that did not have any congregation venues for street children and youth in the sampling frame.

Second, at each of the selected venues, we applied respondent driven sampling (RDS) to recruit the eligible street children and youth for interviews from their network groups. RDS is a chain referral sampling method that produces a stable sample regardless of the make-up of

initial recruits [30]. Since the numbers of the street children and youth during the fixed time interval was small (15 or fewer), we interviewed all eligible street children and youth found in the venue. Overall, we enlisted a total 167 venues in 27 parishes selected from three of five city divisions.

In the absence of census or longitudinal data of locations of street children and youth, we considered the pre-survey mapping exercise as the most feasible and appealing approach to looking at spatial spread of the street children and youth within the city spaces. The mapping exercise was explorative and facilitated the development of the intra-urban mobility aspects in the questionnaire. The lack of neighborhood addresses of specific residences of street children and youth made it difficult to define the movement boundaries. Movements of street children and youth are within city spaces such as streets, markets, bus stops and temporary shelters and therefore preference for a 'residence' cannot be defined by parishes or neighborhoods. Even the interviews were not conducted in their 'homes' but on the streets.

Participants for the key informant interviews were purposively selected and included officials from Kampala Capital City Authority (KCCA), Ministry of Gender, Labour and Social Development and NGO service providers. These individuals were deemed knowledgeable about the urban migration dynamics and behaviour of street children and youth and therefore able to provide the required information. To have an in-depth understanding of the rural-urban migration experiences and behaviours, we conducted in-depth interviews (IDIs) with street children and youth.

### Data collection

Data were collected by 14 trained interviewers (statistician, social scientists, and a child psychologist) using a pre-tested electronic semi-structured questionnaire which was pre-programmed using SurveyCTO on android mobile tablets. The tablets were programmed with internal checks to ensure completeness and logical entries, and consistency of the responses. The identification of the congregation venues/locations and access to street children and youth was made possible with the help of the local urban leaders, service provider NGOs and/or street children landlords/caregivers, referred to as 'Street uncles'. The 'Street uncles' served as local guides and icons of security during the interviewing process. 'Street uncles' is a jargon commonly used by the street children and youth to denote an adult who provides shelter, supervises and monitors street children and youth's activities and to whom the latter must pay allegiance.

To allow full participation, most interviews took place between 09:00 hours and 14:00 hours before the street children and youth were engaged in their street-based activities. Interviews were conducted in the commonly spoken local languages (*Luganda* and *Ngakarimajong*), which ensured a response rate of 100% among the study participants approached. For qualitative interviews, age and participant appropriate topic guides composed of open and closed ended questions were applied. Data from the five (5) key informants (officials from the KCCA and Non-Governmental Organisations (NGOs) and 10 in-depth interviews (6 migrant street youth and 4 caregivers of street children) were collected by the Principal Investigator (PI) and two experienced qualitative research experts in both English and local languages, *Luganda* and *Ngakarimajong*, which are commonly spoken by the street children and youth.

### Measures

The dependent variable, intra-urban mobility was assessed by asking respondents how many places they have moved or lived in since migrating to the city. Within the broader context of our study on which this paper is nested, our questionnaire was designed in such a way that

only street children and youth with a rural-urban migration history responded to the mobility questions. We had no interest in studying the mobility of lifelong native street children and youth. Our study focuses on studying intra-urban mobility of street children and youth, which is different from residential mobility. The definition of residential mobility which is often defined in terms of number of times in the past 12 months residents have moved, is inconsistent in the literature, often paying attention to the spatial aspects of intra-urban mobility [31]. Within the local context, the concept of residential mobility is not applicable given the absence of residential street addresses for residences of the street children and youth in Kampala.

Demographic characteristics of migrant street children and youth included age, marital status, gender, religion, education, schooling status, living arrangements, daily income and rural-urban migration experience. Migration aspects were measured by asking the following questions: *(1) Which district were you born*, *or do you originate from (where your family/parents permanently live)*? Responses to this question included (i) Kampala (ii) other district (specify. . .); *(2) In which year and month did you first come to settle in Kampala*? If the response to question (1) was Kampala or Wakiso district (considered as the Kampala metropolitan area), the respondent skipped subsequent questions that measured the study outcome variable. Therefore, our analysis inevitably excludes street children and youth without a rural-urban migration experience who constitutes only 20% (n = 101) of the overall study sample size.

We then generated the variable "migration status" from questions (1) and (2). Accordingly, street children and youth born outside Kampala metropolitan area and who had migrated to the city were considered as "migrants". These are not immigrants in the actual sense but in-migrants to the city. Migrants were further classified as short-term migrants ($\leq$ 2 years of stay in city) and long-term migrants ($>$ 2 years of stay in city). A recent cohort study done in Uganda used a similar categorisation [32]. Additional aspects on intra-urban mobility were captured during the pre-survey mapping exercise which employed both qualitative interviews and on-site visits to the city spaces.

Behavioural characteristics assessed included risky sexual behaviour and utilisation of sexual and reproductive health (SRH) services. Risky sexual behaviour was assessed by asking the following questions:–*Did you use a condom at last sexual encounter*? *(yes/no). How many sexual partners have you had sexual intercourse with during the past 12 months*?. Risky sexual behaviour was considered if the respondent did not use a condom at last sex and had multiple sexual partners. Use of SRH services in the past 12 months was defined by a "yes" response to all the three questions:—*Have you ever tested for HIV and know status*? *(yes/no); ever used family planning*? *(yes/no); and ever screened for sexually transmitted infections*? *(yes/no)*. Among the sexually active street children and youth, substance abuse was assessed by asking respondents whether they had sexual intercourse under the influence of drugs in the past 12 months (yes/no). Sex work was assessed by asking if the respondent had ever received money or goods in exchange for sex (yes/no).

## Data analysis

Our paper sought to examine the drivers of intra-urban mobility among street children and youth with a rural-urban migration experience who constitute 80.31% (n = 412) of the study sample size. To assess the drivers of intra-urban mobility, we performed the negative binomial regression and other alternative models (Poisson regression and OLS) in STATA (version 15). Our response variable was count data and exhibited over dispersion *(that is, mean = 2.19 vs Variance 3.57)* and hence satisfying the assumptions of the negative binomial regression. Incidence rate ratios (IRR) and 95% confidence intervals (CI) were calculated assuming a negative binomial distribution for the number of intra-urban places lived in since migration. We used

the Bayesian criteria for selection of entering variables in the multivariate model. Predictor variables in the final model include: sex (male/female = 1), age (10–17 vs. 18–24 years = 1), marital status (not married/married/cohabiting = 1), highest education attained (primary/secondary& Tertiary = 1), median shelter occupants (5 and above, 1–4 = 1), daily income (< 1USD/1USD or more = 1), migration status (long/short term migrants = 1), duration of stay in the city (0–2 years = 0, 3–5 years = 1, 6–15 years = 2), sex work (no/yes = 1), use of SRH services (no/yes = 1), risky sexual behaviour (no/yes = 1). The observed decrease in the model number of observations from the actual study sample size could be attributable to the missing values for some variables in the final model which by default, were excluded from our analysis. Collinearity between intra-urban mobility and independent factors was automatically detected. Stratification of the data by age group did not allow any meaningful analysis due to small cell numbers.

All the qualitative interviews were voice recorded, transcribed verbatim and analysed manually using thematic content analysis along predetermined themes [33]. Accuracy of the translations was assured by use of bilingual researchers with experience in conducting and analysing qualitative research with vulnerable children and youth.

### Ethical approval

The study protocol and materials were approved by the research and ethics committees of Maastricht University Faculty of Health, Medicine and Life Sciences, Netherlands (Ref. BvB/KvH/08102018), Makerere University School of Social Sciences, Kampala (Ref. MAKSS REC 12.18.239) and the Uganda National Council of Science and Technology (UNCST) (Ref. HS348ES). Written informed consent and assent were obtained for children aged less than 18 years while consent was obtained from adult participants. As per the UNCST guidelines for research involving humans as research participants, children aged 14–17 years are considered mature and emancipated minors and therefore may independently provide informed consent to participate in research if justified [34]. Assent was obtained from the street children's caregiver/guardian, referred to as 'Street uncles' for street children aged below 18 years and consideration made for illiteracy. Written consent was obtained for participants for the qualitative interviews. Data were analysed anonymously.

## Results and discussion

### Results

Table 1 shows the intra-urban mobility, demographic and behavioural characteristics of the street children and youth stratified by sex and age. Of the 412 street children and youth with a rural-urban migration history, two-thirds (67.23%) had stayed in the city for at least two years, 19.90% for 3–5 years and 12.86% for 6 years or longer. The median duration of stay in the city was 5 years residences (first quartile Q1 = 5 and third quartile Q3 = 14). More than half (54.37%) of the migrant street children and youth made two or more moves within the city spaces.

Intra-urban mobility varied significantly by gender with 68.78% of the males having moved at least two places compared to 37.70% of the females. Although older street youth had more moves compared to the young age group 12–17 years, the observed difference was not statistically significant.

Marital status and daily incomes varied significantly with age and gender. More than a third (39.79%) of the female migrant street children and youth compared to 2.26% of their male counterparts were in conjugal relationships. Less than third (29.48%) of the older migrant street adults were married or cohabiting, with 4.35% engaged in child marriages. Daily earning

**Table 1. Intra-urban mobility, demographic and behavioural characteristics of migrant street children and youth stratified by gender and age, Kampala, Uganda, 2019.**

| Characteristic | All Respondents | Gender | | Age (years) | |
|---|---|---|---|---|---|
| | N (%) | Male n (%) | Female n (%) | 12–17 n (%) | 18–24 n (%) |
| **Intra-urban moves in the city (N = 412)** | | | | | |
| 1 (never moved) | 188(45.63) | 69(31.22) | 119(62.30) ** | 79(49.07) | 109(43.43) |
| 2 | 114(27.67) | 69(31.22) | 45(23.56) | 39(24.22) | 75(29.88) |
| 3 | 53(12.86) | 37(16.74) | 16(8.38) | 28(17.39) | 25(9.96) |
| 4 | 24(5.83) | 20(9.05) | 4(2.09) | 6(3.73) | 18(7.17) |
| 5 to 15 | 33(8.01) | 26(11.76) | 7(3.66) | 9(5.59) | 24(9.56) |
| **Duration of stay in city following migration (N = 412)** | | | | | |
| 0–2 years | 277(67.23) | 132(59.73) | 145(75.92) ** | 125(77.64) | 152(60.56) ** |
| 3–5 years | 82(19.90) | 47(21.27) | 35(18.32) | 24(14.91) | 58(23.11) |
| 6 to 15 years | 53(12.86) | 42(19.00) | 11(5.76) | 12(7.45) | 41(16.33) |
| **Marital status (N = 412) ** | | | | | |
| Married/co-habiting | 81(19.66) | 5(2.26) | 76(39.79) ** | 7(4.35) | 74(29.48) ** |
| Not married | 320(77.67) | 214(96.83) | 106(55.50) | 152(94.41) | 168(66.93) |
| Divorced/separated/widowed | 11(2.67) | 2(0.90) | 9(4.71) | 2(1.24) | 9(3.59) |
| **Highest level of education (N = 326)** | | | | | |
| Primary | 248(76.07) | 142(69.61) | 106(86.89) | 108(88.52) | 140 (68.63) |
| Secondary | 76(23.31) | 61(29.90) | 15(12.30) | 14(11.48) | 62(30.39) |
| Tertiary | 2(0.61) | 1(0.49) | 1(0.82) | 0(0.00) | 2(0.98) |
| **Schooling status (N = 412)** | | | | | |
| Out of school | 377(91.50) | 194(87.78) | 183(95.81) ** | 139(86.34) | 238(94.82) ** |
| In-school | 55(8.50) | 27(12.22) | 8(4.19) | 22(13.66) | 13(5.18) |
| **Religion (N = 412)** | | | | | |
| Non-Christian | 74(17.96) | 67(30.32) | 7(3.66) ** | 36(22.36) | 38(15.14) |
| Christian | 338(82.04) | 154(69.68) | 184(96.34) | 125(77.64) | 213(84.86) |
| **Whom do you mainly stay with? (N = 412)** | | | | | |
| Alone | 51(12.38) | 41(18.55) | 10(5.24) ** | 14(8.70) | 37(14.74) |
| Parents (both or either) | 24(5.82) | 14(6.34) | 10(5.21) | 11(6.83) | 13(4.98) |
| Friends | 224(59.22) | 125(56.56) | 119(62.30) | 104(64.60) | 140(55.78) |
| Siblings/Other Relatives | 83(20.15) | 37(16.74) | 46(24.08) | 32(19.88) | 51(20.32) |
| Spouse/partner | 10(2.43) | 4(1.81) | 6(3.14) | 0(0.00) | 10(3.98) |
| **Daily income (USD) (N = 367)** | | | | | |
| Less than 1 USD | 128(34.88) | 41(21.35) | 87(49.71) ** | 62(43.97) | 66(29.20) ** |
| 1USD or more | 239(65.12) | 151(78.65) | 88(50.29) | 79(56.03) | 160(70.80) |
| **No. of Shelter Occupants (N = 412)** | | | | | |
| 5 and above | 218(52.91) | 74(33.48) | 144(75.39) | 100(62.11) | 118(47.01) ** |
| 1–4 members | 194(47.09) | 147 (66.52) | 47(24.61) | 61(37.89) | 133(52.99) |
| **Involved in sex work (N = 227)** | | | | | |
| No | 209(92.07) | 120(95.24) | 89(88.12) ** | 36(92.31) | 173(92.02) |
| Yes | 18(7.93) | 6(4.76) | 12(11.88) | 3(7.69) | 15(7.98) |
| **Risky sexual behaviour (N = 227)** | | | | | |
| No | 214(94.27) | 118(93.65) | 96(95.05) | 38(97.44) | 176(93.62) |
| Yes | 13(5.73) | 8(6.35) | 5(4.95) | 1(2.56) | 12(6.38) |
| **SRH services use in past 12 months (N = 412)** | | | | | |
| No | 371(90.05) | 194(87.78) | 177(92.67) | 154(95.65) | 217(86.45) ** |
| Yes | 41(9.95) | 27(12.22) | 14(7.33) | 7(4.35) | 34(13.55) |

** Statistically significant at p<0.05.

of at least one USD was significantly higher among the males than the females, and the older youth (18 years and above) earned more than their young counter parts.

Among the sexually active, a small proportion of the street children and youth where involved in sex work and risky sexual behaviour. Although less prevent, sex work was found to be more common among females and older street youth than the male and young street children and varied significantly by age and gender. Access to SRH services was generally low with 41(9.95% (n = 41) having utilised sexually transmitted infections (STIs) screening, family planning (FP) and HIV testing services in the past 12 months. Fewer females than males accessed SRH services. Utilisation of SRH services significantly varied with age, with older street youth (13.55%) having utilised SRH services compared to their young counterparts (4.35%).

Findings from the qualitative interviews showed that street children and youth relocate and move across several city spaces, often between the streets and outside of the streets. Some of them lived on streets while others, especially the girls 'rented' temporary shelters in the informal settlements. Within the city spaces, street children lived mainly in temporary shelters, dilapidated buildings/containers, including staying in shacks in open markets/places and shop verandas. The temporary shelters are governed by 'Street uncles'. 'Street uncles', also called 'Street commanders' are mainly adults who serve as caretakers/landlords of the migrant street children and youth, provide them temporary shelters for accommodation at a fee. They deploy the street children and youth in strategic street locations, including finding them small casual jobs. They monitor the activities of the street children while on the street to ensure that they are safeguarded and return safely to their places of stay and collect "rent" from them for each night spent.

## Demographic and behavioural factors

Table 2 shows findings from the multivariate Poisson regression, negative binomial regression and Ordinary Least Squares (OLS) analyses of intra-urban mobility of street children and youth and predictor variables. Both the Poisson regression (model 1) and negative binomial regression (model 2) yielded similar significant associations between intra-urban mobility and gender, duration of stay, daily income earned and sex work involvement. Notably, the likelihoods of intra-urban mobility were reduced by 30% if the street child or youth was female as opposed to being a male, implying that females were less likely to be mobile than their male counterparts (aIRR = 0.71, 95%CI 0.53–0.96). Migrant street children and youth who had stayed 6 years or longer in the city were 1.5 times more likely to have increased intra-urban mobility than their counterparts with a shorter duration of stay in the city (aIRR = 1.53, 95% CI 1.17–2.01). Migrant street children who earned a daily income of one USD or more were 1.6 times more likely to have increased intra-urban mobility than those who earned less than one USD per day (aIRR = 1.57, 95%CI 1.16–2.13). Migrant street children and youth who were involved in sex work were 1.4 times more likely to have increased intra-urban mobility than youth who did not engage in sex work (aIRR = 1.38, 95%CI 1.01–1.88). The OLS (model 3) did not yield any significant findings and hence not appropriate in perfectly predicting our outcome variable.

**Affordability to pay for rented space.** It is evident that street children and youth's ability to pay for rent to their landlords could influence their intra-urban mobility. Most of them could stay in places for which they were able to pay for accommodation. For instance, in one of the popular sleeping places that the research team visited, we found a group of 200 street children and youth residing in a hostel (one cubical) that was managed by the 'Street uncles. The street children and youth said that hostels were cheaper and safer compared to renting a single room situated within the informal settlements.

**Table 2. Multivariate regression analysis of demographic and behavioural characteristics with intra-urban mobility among migrant street children and youth in Kampala, Uganda, 2019.**

| Variable | (Model 1) | (Model 2) | Model 3 |
|---|---|---|---|
| | Poisson regression aIRR (95% CI) | Negative binomial regression aIRR (95% CI) | (OLS) β (95%CI) |
| Age (18–24 = 1) | 0.94(0.72–1.26) | 0.95(0.71–1.26) | -0.09(-0.77–0.58) |
| Gender (female = 1) | 0.71(0.53–0.96) | 0.71(0.53–0.96) | -0.85(-1.58- -0.13) |
| Marital status (not married = 1) | 1.30(0.95–1.78) | 1.30(0.95–1.78) | 0.74(-0.035–1.52) |
| Level of education attained (secondary + tertiary = 1) | 1.12(0.90–1.39) | 1.12(0.90–1.38) | 0.248 (-0.307–0.804) |
| Median Shelter Occupants (Median and above = 0; Below median = 1) | 1.22(0.96–1.55) | 1.22(0.96–1.55) | 0.44(-0.15–1.03) |
| Daily income earned (>one USD = 1) | 1.57(1.16–2.12) | 1.57(1.16–2.13) | 0.84(0.23–1.45) |
| Involved in sex work (yes = 1) | 1.38(1.01–1.88) | 1.38(1.01–1.88) | 0.89(0.02–1.76) |
| Risky sexual behaviour) (yes = 1) | 1.14(0.85–1.55) | 1.15(0.85–1.55) | 0.36(-0.47–1.19) |
| SRH services use in past 12 months (yes = 1) | 1.01(0.79–1.29) | 1.01(0.79–1.29) | -0.02(-0.645–0.605) |
| Migration status (short term = 1) | 0.99(0.75–1.30) | 0.99(0.75–1.30) | 0.04(-0.687–0.758) |
| Duration of stay in city | | | |
| a.0-2 years (= 0) | 1 | 1 | 1 |
| b.3-5 years (= 1) | 0.99(0.75–1.31) | 0.99(0.75–1.31) | -0.013(-0.675-.647) |
| c.6-15 years (= 2) | 1.52(1.17–2.01) | 1.54(1.17–2.01) | 1.36(0.598–2.12) |

From the qualitative findings, reasons for increased intra-urban mobility among street children and youth identified include

*"The rent paid varies according to the place. A sleeping corridor can go for UGX 500 ($0.15), temporary structure built with mud, grass and tarpaulin at UGX1000 ($0.3), while rent for a room goes for UGX 4000($1.2) for a night"* (**IDI, Street uncle**).

*"Sometimes, private security guards charge us between UGX 1000 to UGX 2000 as 'rent' for sleeping in verandas of shops which they guard at night. If you do not have money to pay, sometimes they can sleep with (rape) you in exchange for space"* (**IDI, Female, Street child**).

**Personal safety.** The kind of reception provided by landlords and safety within the rented spaces was commonly mentioned as another reason that may influence street children and youth's increased mobility within the urban spaces. For personal safety, most of the street children and youth moved or slept in small groups of about five members.

*"We sleep in shifts such that when we notice danger, we can relocate to another street or sleeping place where it's safer"* (**In-depth interview, Street child**)

Most migrant street children and youth said they were likely to stay longer in a city place if they were treated well by their sham landlords. Harassment by the landlords was widely reported as another reason for moving to new places.

*"Where to stay is determined by how much rent you can afford to pay for. Sometimes, when we have money we can pay and have peaceful sleep. Failure to pay for the sleeping space does not only earn you strokes but also the Landlord searches you and takes all the money they find on you. Due to the harassment from the landlords, we must move from one place to another"* (**In-depth interview, Street child**)

**Social support networks.**    We noted that street children and youth with a common ethnicity or same place/district of origin preferred to stay together or move in small groups as a social safety net. The leadership of the migrant street children and youth was found to be clustered around age group, with the older children automatically becoming group leaders.

**Daily income earned.**    Gender and nature of work undertaken by the street children and youth was mentioned as possible drivers for intra-urban mobility. When asked how much a street child or adult earns on a day, one of key informants said:

> *"These street children can earn between UGX 500 and UGX 5000 per day"* **(Key Informant, NGO)**

Common livelihood activities for the street children and youth especially the boys included picking scrap, carrying luggage, fetching water, and offloading trucks while a few of them resorted to pick pocketing. On the other hand, the girls were mainly involved in begging, sorting beans/ground nuts, picking remnants of vegetables from busy markets which they either sold or ate, working in restaurants and vending of merchandise. The street children and youth often moved to busy places where they could easily find causal work, social and monetary support.

## Discussion

This is the first Ugandan study to examine intra-urban mobility and its drivers among rural-urban migrant street children and youth using mixed methods research approaches. We assessed the socio-demographic and behavioural factors associated with intra-urban mobility of street children and youth in Kampala, Uganda. Our findings show that migrant street children and youth, especially the males and older youth, are highly mobile compared to the female and young counter parts. Intra-urban mobility was found to be associated with gender, involvement in sex work, duration of stay in city and a daily earning of one USD or more.

Our study reveals that most migrant street children and youth live under the care of 'Street uncles' who serve as their 'landlords' or 'caretakers' and therefore charge them 'rental fees' for every single night spent. Most places of residence are streets, markets, taxi parks and urban neighbourhoods, especially the slums. Therefore, the propensity of the street children and youth to relocate may be largely predicated by their ability to either pay 'rental' fees and maintain their current shelter or afford the costs of moving to another urban space. Previous studies have linked migrants' ability to pay rent for public housing with a lower average mobility rate [10].

The reduced odds of intra-urban mobility among female street children and youth compared to their male counterparts is a plausible finding. Within the local context, this difference in intra-urban mobility could partly be explained by the fact that, first, girls prefer to stay more solid socially networked groups and highly guarded and safer places. Intra-urban mobility has been found to be dependent upon the cohesion and support among urban migrants [35, 36]. As such, girls are more likely to stay longer in particular locations compared to the boys. This is confirmed by our qualitative findings in which girls were found to be highly guarded or closely monitored by the 'Street uncles' including street night guards. Second, mobility within urban spaces could be influenced by crime. Due to their involvement in gang crimes, male street youth live in a state of fear of being arrested by police and urban authorities [37], which may compel them to change their residences to escape arrests.

The increased likelihood of intra-urban mobility with duration of stay in the city following migration confirms the cultural learning model, which presupposes that adaptation occurs

through migrants' learning of culture-specific skills that would enable them to negotiate their ways in the new cultural environment [38, 39]. To this end, it is reasonable to assume that older migrant street children and youth who have stayed in the city for 6 or longer have fully adapted to the new street environment. Therefore, they are likely to have the ability and strength to easily navigate the complex urban environment than the newcomers who may take some time to establish social and peer support networks before making several moves. Also, it has been argued that the inherent mobility in hostile urban environments can potentially lead young men to assume multiple and perhaps contradictory identities in order to survive the vagaries of urban life [40].

The association of the daily income above the poverty threshold of one USD daily and intra-urban mobility shows that an increase in daily income is a motivation for street children and youth's intra-urban mobility. This result suggests that street children and youth may choose to move to strategic urban locations that offer them better casual work opportunities and best match their payment abilities for rent and to meet basic needs. Within the local context of the study, most migrant street children and youth come from underprivileged rural areas and households that are struggling to meet basic needs. As such, migrant street children and youth are often perceived as breadwinners and therefore work hard to meet their basic needs and those of the families left behind. Previous work has shown that most residential movers often come from economically disadvantaged families [41, 42].

Street youth involved in sex work were twice as likely to move multiple places within the city spaces as those who were not engaged in sex work. This finding is not surprising as sex worker mobility may be closely linked to livelihoods, which involves finding casual work in different city locations such as bars, parks, markets and bus stops. Previous studies show that females who move have been found to have increased rates of premarital sex and teenage pregnancy than females who remained residentially stable [43, 44].

This study has several strengths. The focus on intra-urban mobility of 412 migrant street children and youth aged 12–24 years is one of the important strengths. We were able to examine the drivers of intra-urban mobility of migrant street children and youth across a wide range of demographic and behavioural measures. Inclusion of known co-variates such as age and education into the final model based on previous literature also increases the strength of the study. As a final note on the analysis, several common statistical techniques of the generalised linear models, including the usual form of regression analysis, Ordinary Least Squares regression, and binomial logit modelling have been used to model migration [45]. While the standard Poisson and logistic regression models have been applied to model determinants of intra-urban residential mobility [46, 47], in our study, both the standard Poisson and negative binomial regression models perfectly predicted the outcome count variable, with the negative binomial regression being the most appropriate model for assessing the predictors of intra-urban mobility.

However, the study had some limitations that could be addressed in future research. First, our data are cross-sectional and thus preclude our ability to determine the direction of causality. We also have no information about movement of family members of the street children, which may be an important correlate of their residential mobility. While intra-urban mobility is clearly measured in this study, the measure does not account for the distances moved and times that the street children and youth spent at each location. Within the local context, the absence of clearly demarcated residences, neighbourhoods and streets without assigned physical addresses of street children and youth as well as the lack of an existing longitudinal database of all urban residents including the migrant street children and youth renders it difficult to compare movements in terms of the short distances moved across the streets and/or urban spaces. Even so, Simmons (1968) asserts that location is often incidental in the decision-

making process and most moves cover only short distances [48]. Third, recall and social desirability predispositions could have influenced responses to questions regarding intra-urban migration experiences and sexual and reproductive practices, respectively. Lastly, our analysis is based on a subset of migrant street children and youth which limits generalisability of the study findings.

Despite the above limitations, our findings could be reliable due to the application of a large and representative sample of street children and youth and the mixed methods that reinforced each other. In developed countries, residential mobility studies are often modelled on longitudinal or census data, which is not feasible within our local context [49, 50]. This paper not only adds value to the current understanding of the structural drivers of intra-urban mobility in low-income settings, particularly sex work but also provides a stimulus for future research on diverse drivers of intra-urban mobility of the street children and youth to the fullest.

## Conclusion

In conclusion, our study shows that migrant street children and youth are highly intra-urbanly mobile, with patterns that differ by demographic and structural factors. The drivers of intra-urban mobility among migrant street children and youth include daily income earned, gender, duration of stay, involvement in sex work and personal safety within the city spaces. The study further demonstrates a considerable degree of vulnerability associated with mobility between urban spaces which is also characterized by long-term urban residents and sham landlords who exploit the migrant street children and youth.

Our findings suggest the need for urban housing and health policies and programs for migrant street children and youth to take into account their intra-urban mobility patterns and its drivers. We acknowledge that intra-urban mobility of street children and youth is not a phenomenon to be solved but a reality to be managed. Effective urban programs should be guided by the whole-of-government and whole-of-society inclusive approaches and developed in the best interest of the street child. The government of Uganda and KCCA should consider safe and cheaper shelter/housing options and targeted policy interventions for the street children and youth, enact policies to safeguard them from all forms of violence including sexual exploitation, and regulate sex work in the city.

Economic strengthening through integral entrepreneurship skills could help reduce the level of sexual exploitation among the street children. The gender differences in intra-urban mobility among street children and youth call for the need to enact gender responsive policy interventions to address intra-urban mobility gender-associated risks and vulnerabilities. Finally, the lack of reliable census data on street children and youth's intra-urban mobility and addresses of their locations, necessitates the need for establishing a longitudinal database by the KCCA and National Statistics Bureau to profile intra-urban mobility patterns of all rural-urban migrants to facilitate future research.

## Supporting information

**S1 Dataset.**
(ZIP)

## Acknowledgments

Special thanks go to the research team for their invaluable support during implementation of the study. The Applied Research Bureau (ARB), Kampala, Uganda supported the training of

the research assistants. We are grateful to the street children and youth and other stakeholders who volunteered to participate in this study.

## Author Contributions

**Conceptualization:** Mulekya Francis Bwambale, Paul Bukuluki, Cheryl A. Moyer, Bart H. W. van den Borne.

**Formal analysis:** Mulekya Francis Bwambale.

**Funding acquisition:** Mulekya Francis Bwambale, Paul Bukuluki, Bart H. W. van den Borne.

**Investigation:** Mulekya Francis Bwambale.

**Methodology:** Mulekya Francis Bwambale, Paul Bukuluki, Cheryl A. Moyer, Bart H. W. van den Borne.

**Project administration:** Mulekya Francis Bwambale.

**Resources:** Mulekya Francis Bwambale, Paul Bukuluki, Cheryl A. Moyer.

**Supervision:** Paul Bukuluki, Cheryl A. Moyer, Bart H. W. van den Borne.

**Writing – original draft:** Mulekya Francis Bwambale.

**Writing – review & editing:** Mulekya Francis Bwambale, Paul Bukuluki, Cheryl A. Moyer, Bart H. W. van den Borne.

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
