## [Decision Letter · Decision Letter 0]

16 Apr 2020

PONE-D-19-29260

Predictors of Intra-Urban Residential Mobility of Street Children in Uganda: Implications for policy and practice

PLOS ONE

Dear Dr. Mulekya-Bwambale,

Thank you for submitting your work to Plos ONE and apologies for the relatively long time needed to send you the Journal’s feedback. To make a long story short, the paper required a rather lengthy period of time to obtain at least two referee reports, that provided two rather opposite views on your paper. Referee 1 suggests to reject it, while Referee 2 advises minor revisions. After waiting for a third reviewer to somewhat settle this issue I decided to make a final decision myself in order to speed up the process.

After carefully reading your paper and the two reviews I am attaching, I see some potential in it but I do agree with many of the points raised by Referee 1. While also suggesting you to go through all points raised by both reviewers in the attached reports, let me here briefly recap the major points you should in ,my view address in order to convince both reviewers, and especially Reviewer 1:

Please provide details on the way mobility is defined in your data sample. Please control for the the exposure time in your analyses; explain why non-migrants have been excluded from your analyses as a potential control group; how you define mobility within Kampala; and, lastly, what are the physical boundaries describing this movement.Please explain why the logistic model is discarded from your analyses and why in the first place you decide to retain it and show the results despite its worse performance.Please explain why you are left with 207 observations out of the original 513 being interviewed. What are the criteria that excludes them? Does this introduce a bias in your estimates?Please clarify whether the data you are analysing will be made available.Please review your text so as to better clarify the aim of your work from the outset and strengthen its logic.Please review your definition of “Street children”, that is used in your paper to talk about 12-24 year old people, while a usual definition of a child is arguably under 18.

Please prepare a reply letter addressing each point raised by both reviewers in terms of the way you believe they have, or haven’t, been solved in your revised paper.

I look forward to reviewing your revised paper in due time and I thank you once again for thinking of Plos ONE as a possible outlet for your research.

Kind regards,

Andrea Caragliu

We would appreciate receiving your revised manuscript by May 31 2020 11:59PM. To enhance the reproducibility of your results, we recommend that if applicable you deposit your laboratory protocols in protocols.io, where a protocol can be assigned its own identifier (DOI) such that it can be cited independently in the future. For instructions see: http://journals.plos.org/plosone/s/submission-guidelines#loc-laboratory-protocols

We look forward to receiving your revised manuscript.

Kind regards,

Andrea Antonio Guido Caragliu

Academic Editor

PLOS ONE

Journal Requirements:

3. We note that Figures 1 and 2in your submission contain map images which may be copyrighted. All PLOS content is published under the Creative Commons Attribution License (CC BY 4.0), which means that the manuscript, images, and Supporting Information files will be freely available online, and any third party is permitted to access, download, copy, distribute, and use these materials in any way, even commercially, with proper attribution. For these reasons, we cannot publish previously copyrighted maps or satellite images created using proprietary data, such as Google software (Google Maps, Street View, and Earth). For more information, see our copyright guidelines: http://journals.plos.org/plosone/s/licenses-and-copyright.

a) You may seek permission from the original copyright holder of Figure(s) [#] to publish the content specifically under the CC BY 4.0 license.

b).    If you are unable to obtain permission from the original copyright holder to publish these figures under the CC BY 4.0 license or if the copyright holder’s requirements are incompatible with the CC BY 4.0 license, please either i) remove the figure or ii) supply a replacement figure that complies with the CC BY 4.0 license. Please check copyright information on all replacement figures and update the figure caption with source information. If applicable, please specify in the figure caption text when a figure is similar but not identical to the original image and is therefore for illustrative purposes only.

Additional Editor Comments (if provided):

Decision letter PONE-D-19-29260

"Predictors of Intra-Urban Residential Mobility of Street Children in Uganda: Implications for policy and practice"

Dear Dr. Mulekya-Bwambale,

Thank you for submitting your work to Plos ONE and apologies for the relatively long time needed to send you the Journal’s feedback. To make a long story short, the paper required a rather lengthy period of time to obtain at least two referee reports, that provided two rather opposite views on your paper. Referee 1 suggests to reject it, while Referee 2 advises minor revisions. After waiting for a third reviewer to somewhat settle this issue I decided to make a final decision myself in order to speed up the process.

After carefully reading your paper and the two reviews I am attaching, I see some potential in it but I do agree with many of the points raised by Referee 1. While also suggesting you to go through all points raised by both reviewers in the attached reports, let me here briefly recap the major points you should in ,my view address in order to convince both reviewers, and especially Reviewer 1:

1. Please provide details on the way mobility is defined in your data sample. Please control for the the exposure time in your analyses; explain why non-migrants have been excluded from your analyses as a potential control group; how you define mobility within Kampala; and, lastly, what are the physical boundaries describing this movement.

2. Please explain why the logistic model is discarded from your analyses and why in the first place you decide to retain it and show the results despite its worse performance.

3. Please explain why you are left with 207 observations out of the original 513 being interviewed. What are the criteria that excludes them? Does this introduce a bias in your estimates?

4. Please clarify whether the data you are analysing will be made available.

5. Please review your text so as to better clarify the aim of your work from the outset and strengthen its logic.

6. Please review your definition of “Street children”, that is used in your paper to talk about 12-24 year old people, while a usual definition of a child is arguably under 18.

Please prepare a reply letter addressing each point raised by both reviewers in terms of the way you believe they have, or haven’t, been solved in your revised paper.

I look forward to reviewing your revised paper in due time and I thank you once again for thinking of Plos ONE as a possible outlet for your research.

Kind regards,

Andrea Caragliu

Reviewers' comments:

Reviewer's Responses to Questions

**Comments to the Author**

1. Is the manuscript technically sound, and do the data support the conclusions?

Reviewer #1: No

Reviewer #2: Yes

2. Has the statistical analysis been performed appropriately and rigorously? 

Reviewer #1: No

Reviewer #2: Yes

3. Have the authors made all data underlying the findings in their manuscript fully available?

Reviewer #1: No

Reviewer #2: Yes

4. Is the manuscript presented in an intelligible fashion and written in standard English?

Reviewer #1: No

Reviewer #2: Yes

5. Review Comments to the Author

Reviewer #1: • Is the manuscript technically sound? Has the statistical analysis been performed appropriately and rigorously?

This paper addresses two important and under-studied topics in low- middle-income countries, namely intra-urban mobility, and street children. However, the execution of analysis is flawed. Firstly, the definition of mobility is problematic since the period observed (the exposure time) differs from person to person when the outcome is defined as the number of places stayed in since migrating to the city. You can’t then compare the number of moves. Also, the paper does not do enough to address these children as being a select group- since, from my understanding, they are all migrants to the city. Why are the non-migrants excluded? Indeed, clarity in terms of rural-urban and intra-urban migration is needed throughout. It is also unclear what is defined as mobility within Kampala. Is someone considered to have moved if they shift parish? Or a certain distance threshold? What are the physical boundaries to describe this movement?

Secondly, although the paper relies on what seems to be a unique set of data on street children, the data is not exploited to understand the drivers of their mobility to the fullest. The modelling relies on two methods (logistic and poisson) but the authors reject the logistic method. So one wonders why they even show the results of this model. The model is based on 207 observations but 513 were interviewed. What happened to the rest of the street-children in the model? (Only migrants were modelled it seems- but why?) Are they missing some answers to some questions? What are the criteria that excludes them- and does this cause bias?

• Is the data fully available?

There is no mention of the data availability.

• Is the manuscript presented in intelligible fashion and written in standard English?

The introduction section requires a fair bit of work- in part it is repetitious, and the ideas do not flow clearly from one statement to the next. It should also be expanded to cover some of the key relationships examined. For example, in the introduction we are not given enough to understand the relationship between having safe sex and residential mobility.

Some language editing is needed and general re-reading of text. For example, on page 13, 2nd line the text reads “earn between UGX 5000 to UGX 5000 per day”. Another example- in Table 4 on page 16, there are two variables (categories) defined exactly the same “Duration of stay (<2 years)”.

An additional problem with the paper is that, although the paper includes a long list of references – they are not accurately used. I noted a few that were misplaced and that raises doubts as to whether the rest of the citations are correct.

Minor notes:

Street children is used here to refer to 12-24 year olds. This is a bit misleading as the definition of a child is arguably under 18 years old – if not 16. Using the term street children is therefore confusing. The paper refers to youth and perhaps “street youth” could be a better and more accurate term.

There is likely bias in the responses to questions on sex and use of reproductive health services. These are sensitive topics and “true” responses are not always disclosed to the interviewers. At minimum comment on this should be included.

Not much is gained from the bivariate analysis in Table 2, beyond that of the multivariate modelling.

In the results section it is noted where the street children reside more (the neighborhoods which are mainly slums)- Figure 2. Yet not all neighbourhoods were included in the sampling. It is misguided to define the location of street children as findings when parishes were pre-selected for possibility of contacting street children. A more appealing approach to looking at the spatial spread of children would be to see where they move from and where to. This could also help in understanding the drivers of their mobility. They may move to parish where health facilities are open to seeing street children. Or from parishes where gangs are particularly violent.

Overall, this paper aims to use unique data on street children but falls short. I would suggest to authors to think what would be a more beneficial way to use the data. Perhaps it is not even the street children's mobility that needs to be examined as the outcome. Attention to this vulnerble under-studied population is commended.

Reviewer #2: General comments:

Thanks for the opportunity to review this interesting article on an understudied topic in low income countries. Generally this is a well conducted study and well written. My comments are mainly to address clarity with statistical approaches and the interpretation of findings/conclusions. I will point out each of these in the relevant sections

1- Title: I feel the word predictor is too strong to refer to associations that are derived from a cross sectional study. I would propose “Intra-urban mobility and associated factors among street children in Kampala, Uganda” with emphasis on Kampala because this was not a country wide study but was focused on one city. The findings from this city may not necessarily be generalizable to all cities in the country.

Abstract:

2- Replace degree with “frequency” the second sentence to We evaluated both the frequency of residential mobility and characteristics “

Introduction

3- The first sentence should be supported with a citation, if there is evidence that urban street children are on the increase.

4- “Street children have been defined as any individuals for whom the street (including unoccupied dwellings) have become their place of living and/or source of livelihood, inadequately protected and supervised by responsible adults” Does this definition include adults? If not you need to clarify in the background why you called people aged 18-24 years street children. Otherwise we may be broadly referring to homeless people but not street children. If there is a strong reason for including young adults this should be made clearer in the background.

Methods:

5- What is the rationale for sampling a wide age group 12-24 years? Unless you are including adults who first migrated as children and then transitioned into adults on the streets. Otherwise by the legal definition of a child in Uganda, you should have limited to age < 18 years.

6- According to Uganda National Council for Science and Technology (UNCST) guidelines, assent in conjunction with parental consent is needed for minors but children age 14-17 years who are “emancipated minors” because they are able to fend for themselves can give their own consent. Could you clarify how you obtained parental consent for street children under 14 years and therefore included these children in the study?

7- Where is it is clear as to how the 27 parishes were selected, It is not clear how the three divisions were selected out of five. Please clarify.

8- The use of venue based-time space sampling is on spot. However, it would be good to describe who the key informants were and how they were identified. This may indicate if adequate mapping of venues was conducted. Was it possible for the data collectors to locate these venues themselves in some instances? Were street children involved in locating other venues?

9- Figure 1 is rather blurred. There is need to increase the resolution. Otherwise it is a nice map except for the ineligible

10- About the questionnaire, did you develop this from scratch or did you adapt a standardized questionnaire from other settings?

11- Children were accessed through street uncles. Could you clarify who identified the street uncles since the uncles seem to be in some form of illegal business, it is important to highlight how they were identified. This would perhaps also help in suggesting future recommendations.

12- Outcome measurement: I have a particular concern regarding the measurement of the main outcome. Since this was defined as number of places the children lived in, there may be major differences in recall for recent migrant versus old migrants especially when they lived in several places. It would have been more accurate to limit the period of recall to the last two years or less. I imagine a study participant who is 24, and migrated 12 years ago, may have problems with recall and this recall period differs depending on how long they lived on the street.

13- Independent variables: A key behavioral variable I expected was drug use, as this could influence sexual behavior. Is there any reason why data on this variable was not collected?

14- Data Analysis: The rationale of using both continuous and categorical outcomes for residential mobility needs to be clarified. From my view, since the outcome was common (50% residential mobility, odds ratios should not have been computed. It is recommended that for a common outcome (>25% prevalence), odds ratios tend to exaggerate associations, as logistic regression is best used for rare outcomes. In such a scenario, Poisson regression is recommended even for a categorical outcome. But in this case since you also considered residential mobility as a continuous outcome, there was no need to also confirm this by logistic regression which is already known to exaggerate associations when an outcome is common. Indeed you reported that some associations disappeared or were attenuated when you treated the outcome as continuous. I suggest you drop the logistic regression in your analysis and only keep the continuous outcome. The aim of this paper is not to compare which approach is better, as this is already known. Just by looking at how frequent the outcome was, a decision should be reached on what approach to use.

15- Ethics: Please clarify this statement: ”Written informed consent and assent were obtained for all participating subjects”. Were all the children literate to give a written consent? Were there instances when witnesses were needed? What about the minors, how did you get parents/guardians to give consent? Could you also clarify what steps you took regarding those identified with STIs or other illnesses?

Results

16- Table 1; this is a very nice table separating the children from the adults. It brings out clearly who is a child (12-17) and an adult (18-24) and I would recommend keeping these groups separate in all analyses. We can clearly see these are different groups eg. Although I know that that stratifying these groups will cause small numbers. But this may be something to do with the design of the study. You rapidly recruited the sample of 500 within 2 months. It would have been good to recruit more of the younger group referred to as children. Considering the high prevalence of residential mobility, the sample size may still be adequate for separate analysis. I propose you conduct a sub-analyses to explore the associations for the different age groups.

Table 1 should include both demographic and behavioral characteristics. For example gender differences in sexual behaviour such as involvement in commercial sex are masked in the other tables but table 1 could ideally show this.

17- What is the difference between the two maps in figure 1 and 2? They seem to be communicating the same message. Keeping one figure may be sufficient.

18- Reporting qualitative findings: The glue between quantitative findings and qualitative lacking. Could you preferably use sub-headings based on themes emerging from qualitative findings, and then reinforce with findings from the quantitative

19- Tables 2 and 3 with the associated text could be excluded based on arguments highlighted before.

Discussion

The discussion is well structured but I suggest that the emphasis should be on factors that the authors should not focus on the advantages of using poission over logistic regression since the decision to use the former should have been taken at analytical stage based on the frequency of the outcome measure. So this statement “Application of the Poisson regression model as an alternative to logistic regression is another strength. In our analysis, logistic regression yielded high odds compared to the Poisson regression model, with SRH service use as significant predictor of residential mobility. However, when Poisson regression model was performed, the relationship become insignificant, implying that the Poisson model is a more appropriate for correctly identifying the predictors for number of street children residential places moved while controlling for confounding.” And other associated statements regarding the analytical approach should be omitted. Any conclusions drawn based on the logistic regression should be deleted as it was not the appropriate approach to analysis.

On page 18, in the statement; “Notably, involvement in sex work was associated with street children increased residential mobility, with more males than females having engaged in sex in exchange for money”, the comparison of males and females is not shown in the data and therefore speculative. Please add this in the results section if it exists.

6. PLOS authors have the option to publish the peer review history of their article (what does this mean?). If published, this will include your full peer review and any attached files.

Reviewer #1: No

Reviewer #2: Yes: Gershim Asiki

---

## [Author Response · Author response to Decision Letter 0]

22 Jun 2020

Additional Editor Comments (if provided):

Decision letter PONE-D-19-29260

"Predictors of Intra-Urban Residential Mobility of Street Children in Uganda: Implications for policy and practice"

Dear Dr. Mulekya-Bwambale,

Thank you for submitting your work to Plos ONE and apologies for the relatively long time needed to send you the Journal’s feedback. To make a long story short, the paper required a rather lengthy period of time to obtain at least two referee reports, that provided two rather opposite views on your paper. Referee 1 suggests to reject it, while Referee 2 advises minor revisions. After waiting for a third reviewer to somewhat settle this issue I decided to make a final decision myself in order to speed up the process.

After carefully reading your paper and the two reviews I am attaching, I see some potential in it but I do agree with many of the points raised by Referee 1. While also suggesting you to go through all points raised by both reviewers in the attached reports, let me here briefly recap the major points you should in ,my view address in order to convince both reviewers, and especially Reviewer 1:

1. Please provide details on the way mobility is defined in your data sample. Please control for the the exposure time in your analyses; explain why non-migrants have been excluded from your analyses as a potential control group; how you define mobility within Kampala; and, lastly, what are the physical boundaries describing this movement.

Response: An explanation of the rationale for the sub-analysis of migrant street children has been included in the abstract, introduction and methods sections. The main interest of the study is on understanding mobility of migrant street children and not lifelong natives of Kampala city. We hypothesise that migrant street young people are likely to experience disproportionate intra-urban mobility compared to the lifelong or native street children and youth who may have gained residential stability over time. 

On the issue of physical boundaries, the lack of addresses of specific locations of places of residence and streets in the city makes it difficult to define boundaries. Movements of street children are within city spaces such as streets, markets, and temporary shelters, and preference for a “residence’ cannot be defined by parishes. Even so, the interviews were not conducted in their ‘homes’ but on the streets. 

2. Please explain why the logistic model is discarded from your analyses and why in the first place you decide to retain it and show the results despite its worse performance.

Response: We repeated the analysis as advised by both Reviewers and adopted the poison regression model. Binary logistic regression has been dropped. 

3. Please explain why you are left with 207 observations out of the original 513 being interviewed. What are the criteria that excludes them? Does this introduce a bias in your estimates?

Response: The difference between model number of observations (n=207) and the actual sample size of migrant street children could be attributable to the missing values for some variables in the model which by default are excluded from our analysis. Our interest of this study is the migrant street children. By design of the tool, we can only analyse for the migrant street children who responded to the question that measures intra-city mobility based on number of places stayed since migrating to Kampala. It also important to note that only 20% of the street children were born in Kampala metropolitan area (non-migrants). 

4. Please clarify whether the data you are analysing will be made available.

Response: Yes, it is available upon request. 

5. Please review your text so as to better clarify the aim of your work from the outset and strengthen its logic.

Response: The manuscript has gone through extensive review and revision. 

6. Please review your definition of “Street children”, that is used in your paper to talk about 12-24 year old people, while a usual definition of a child is arguably under 18.

Response: This has been addressed in the revised manuscript – introduction and methods. 

7. Please prepare a reply letter addressing each point raised by both reviewers in terms of the way you believe they have, or haven’t, been solved in your revised paper.

Response: Done 

Response: This has been done. 

I look forward to reviewing your revised paper in due time and I thank you once again for thinking of Plos ONE as a possible outlet for your research.

Kind regards,

Andrea Caragliu

 

Reviewers' comments:

Reviewer's Responses to Questions

Comments to the Author

1. Is the manuscript technically sound, and do the data support the conclusions?

Reviewer #1: No

Reviewer #2: Yes

Authors’ Response to Reviewer 1: We believe our findings and conclusion are fully supported by the data and matching analysis. 

2. Has the statistical analysis been performed appropriately and rigorously? 

Reviewer #1: No

Reviewer #2: Yes

 Authors’ Response to Reviewer 1: The data have been further interrogated to address the above concerns. The binary logistic model has been dropped. Additional demographic and behavioral drivers have included in Table 1 as suggested. 

3. Have the authors made all data underlying the findings in their manuscript fully available?

Reviewer #1: No

Reviewer #2: Yes

Authors’ Response to Reviewer 1: Our data are available upon request. 

4. Is the manuscript presented in an intelligible fashion and written in standard English?

Reviewer #1: No

Reviewer #2: Yes

Authors’ Response to Reviewer 1: This manuscript has gone through extensive review and revision.

5. Review Comments to the Author

Please use the space provided to explain your answers to the questions above. You may also include additional comments for the author, including concerns about dual publication, research ethics, or publication ethics. (Please upload your review as an attachment if it exceeds 20,000 characters).

Reviewer #1: • Is the manuscript technically sound? Has the statistical analysis been performed appropriately and rigorously?

This paper addresses two important and under-studied topics in low- middle-income countries, namely intra-urban mobility, and street children. However, the execution of analysis is flawed. 

Response: 

We have addressed all the gaps in the analysis that we consider critical to this paper. Details are shown in the methods. We have restricted the sub-analysis to data of migrant street children and adapted Poisson regression. Finally, we have expounded on the local context of the study in all sections to enhance understanding of our study.

Firstly, the definition of mobility is problematic since the period observed (the exposure time) differs from person to person when the outcome is defined as the number of places stayed in since migrating to the city. You can’t then compare the number of moves. Also, the paper does not do enough to address these children as being a select group- since, from my understanding, they are all migrants to the city. Why are the non-migrants excluded? Indeed, clarity in terms of rural-urban and intra-urban migration is needed throughout. It is also unclear what is defined as mobility within Kampala. Is someone considered to have moved if they shift parish? Or a certain distance threshold? What are the physical boundaries to describe this movement?

Response: 

An explanation of the rationale for the sub-analysis of migrant street children has been provided in the abstract, introduction and methods sections. 

The main interest of the study is on understanding mobility of migrant street children and not mobility of lifelong natives of Kampala city. We hypothesise that migrant street young people are likely to experience disproportionate intra-urban mobility compared to the lifelong or native street children and youth who may have gained residential stability over time. 

Within the local contexts, intra-urban mobility could not be defined by parishes due to lack of addresses. We used the GIS coordinates to map the congregation areas by parish, but the maps seemed to make little sense to the readers. Movements are between streets and involve short distances within the city spaces. We reflect the omission of distance as a limitation. 

We defined migrants by ‘place/district of origin’ that is different from Kampala city and we then further categorized migrant street children as long term (> 2 years) or short term ( ≤ 2 years) migrants based on the duration of stay since moving to the city. The questionnaire was structured in a way that non-migrants did not respond to the question that measures intra-urban mobility. In our study, only 20% of street children were born in Kampala city and thus considered lifelong native/non-migrant street children. 

This being a non-longitudinal study, we believe that the duration of stay since migrating to the city reflects the exposure time aspect. We further expound on the different dimensions of intra-mobility by previous studies in the introduction section. 

Secondly, although the paper relies on what seems to be a unique set of data on street children, the data is not exploited to understand the drivers of their mobility to the fullest. The modelling relies on two methods (logistic and poisson) but the authors reject the logistic method. So one wonders why they even show the results of this model. The model is based on 207 observations but 513 were interviewed. What happened to the rest of the street-children in the model? (Only migrants were modelled it seems- but why?) Are they missing some answers to some questions? What are the criteria that excludes them- and does this cause bias?

Response: 

We have dropped the binary logistic regression model and maintained the Poission model. We have included the details on sub-analysis in the methodology section of the paper. We provide details of the questions in the introduction and methods section on measures. 

The difference between model number of observations (n=207) and the actual sample size of migrant street children could be attributable to the missing values for some variables in the model which by default are excluded from our analysis. 

We have considered your suggestion on exploring drivers of intra-urban mobility to the fullest as recommendation for future research. 

• Is the data fully available? 

There is no mention of the data availability.

Response: Data is available upon request. 

• Is the manuscript presented in intelligible fashion and written in standard English?

The introduction section requires a fair bit of work- in part it is repetitious, and the ideas do not flow clearly from one statement to the next. It should also be expanded to cover some of the key relationships examined. For example, in the introduction we are not given enough to understand the relationship between having safe sex and residential mobility.

Some language editing is needed and general re-reading of text. For example, on page 13, 2nd line the text reads “earn between UGX 5000 to UGX 5000 per day”. Another example- in Table 4 on page 16, there are two variables (categories) defined exactly the same “Duration of stay (<2 years)”.

Response: 

We have addressed the textual issues in the revised manuscript to ensure logical flow and reworked some of the wording. We have expanded the introduction to cover the scanty literature sexual behaviour and mobility. 

An additional problem with the paper is that, although the paper includes a long list of references – they are not accurately used. I noted a few that were misplaced and that raises doubts as to whether the rest of the citations are correct.

Reference: References have been adjusted accordingly. 

Minor notes:

Street children is used here to refer to 12-24 year olds. This is a bit misleading as the definition of a child is arguably under 18 years old – if not 16. Using the term street children is therefore confusing. The paper refers to youth and perhaps “street youth” could be a better and more accurate term.

Response: 

While the legal definition of a child in Uganda is any person aged less than 18 years, there is no legal definition of a street child. We have changed the study population to read “street children and youth’’ as suggested although, by Ugandan law, a youth is any person aged 15 to 35 years. This detail is included in the introduction. 

There is likely bias in the responses to questions on sex and use of reproductive health services. These are sensitive topics and “true” responses are not always disclosed to the interviewers. At minimum comment on this should be included.

Response: Captured in the limitations. 

Not much is gained from the bivariate analysis in Table 2, beyond that of the multivariate modelling.

Response: 

Table 2 has been deleted and replaced by a new table of findings from the Poisson regression – captures both the bivariate (model 1) and multi-variate (model 2) analysis findings. 

In the results section it is noted where the street children reside more (the neighborhoods which are mainly slums)- Figure 2. Yet not all neighbourhoods were included in the sampling. It is misguided to define the location of street children as findings when parishes were pre-selected for possibility of contacting street children. A more appealing approach to looking at the spatial spread of children would be to see where they move from and where to. This could also help in understanding the drivers of their mobility. They may move to parish where health facilities are open to seeing street children. Or from parishes where gangs are particularly violent.

Response: 

Both maps (figures 1& 2) have been scrapped. 

Kampala city does not have clearly demarcated neighborhoods nor addresses of streets and locations where street children congregate. In this regard, we focus the study on intra-urban mobility between city spaces and not residences per se. In the absence of any mapping or longitudinal data on city neighborhoods and congregation areas for street children, we thought a pre-survey mapping exercise that adopted qualitative approaches was the most feasible and appealing approach to appreciating the spatial spread of children and hence informing our sampling strategy. We did not include parishes that were found not to have any street children congregation venues in our sample. We have elaborated in our sampling strategy how parishes and venues were selected.

Overall, this paper aims to use unique data on street children but falls short. I would suggest to authors to think what would be a more beneficial way to use the data. Perhaps it is not even the street children's mobility that needs to be examined as the outcome. Attention to this vulnerble under-studied population is commended.

Response: 

We acknowledge that this topic is not well researched. The study is of interest to scholars and stakeholders with a passion for intra-urban migration of rural-urban homeless youth living in cities and more so, it remains beneficial to the Kampala Capacity City Authorities and the Government of Uganda. We affirm that our study is exceptional, and the selected outcome is suitable for this scholarly piece of work given the local context. Future publications to be generated from our data could consider other outcomes of interest. 

Reviewer #2: General comments:

Thanks for the opportunity to review this interesting article on an understudied topic in low income countries. Generally this is a well conducted study and well written. My comments are mainly to address clarity with statistical approaches and the interpretation of findings/conclusions. I will point out each of these in the relevant sections

1- Title: I feel the word predictor is too strong to refer to associations that are derived from a cross sectional study. I would propose “Intra-urban mobility and associated factors among street children in Kampala, Uganda” with emphasis on Kampala because this was not a country wide study but was focused on one city. The findings from this city may not necessarily be generalizable to all cities in the country.

Response: 

Title has been changed to read “Demographic and Behavioural Drivers of Intra-urban mobility of migrant Street Children and Youth in Kampala, Uganda”

Abstract:

2- Replace degree with “frequency” the second sentence to We evaluated both the frequency of residential mobility and characteristics “

Response: Comment has been addressed in the abstract and narrative manuscript. 

Introduction

3- The first sentence should be supported with a citation, if there is evidence that urban street children are on the increase.

Response: 

Appropriate citations have been included following the changes. 

4- “Street children have been defined as any individuals for whom the street (including unoccupied dwellings) have become their place of living and/or source of livelihood, inadequately protected and supervised by responsible adults” Does this definition include adults? If not you need to clarify in the background why you called people aged 18-24 years street children. Otherwise we may be broadly referring to homeless people but not street children. If there is a strong reason for including young adults this should be made clearer in the background.

Response: 

The terminology has been revised to read “street children and youth” throughout the manuscript. 

Methods:

5- What is the rationale for sampling a wide age group 12-24 years? Unless you are including adults who first migrated as children and then transitioned into adults on the streets. Otherwise by the legal definition of a child in Uganda, you should have limited to age < 18 years.

Response: 

The operational definition has been amended to suit street children and youth. We think this embraces the study age group 12 to 24 years. 

6- According to Uganda National Council for Science and Technology (UNCST) guidelines, assent in conjunction with parental consent is needed for minors but children age 14-17 years who are “emancipated minors” because they are able to fend for themselves can give their own consent. Could you clarify how you obtained parental consent for street children under 14 years and therefore included these children in the study?

Response: 

Under the ethics section, we have provided details and clarifications regarding informed consent for minors. 

7- Where is it is clear as to how the 27 parishes were selected, It is not clear how the three divisions were selected out of five. Please clarify.

Response: 

Guided by the mapping data, we used both random and proportionate stratified sampling to enlist parishes and venues. Thus, parishes with more congregation venues had proportionately a high sample of street children interviewed. This is mentioned in the sampling strategy under methods section. 

8- The use of venue based-time space sampling is on spot. However, it would be good to describe who the key informants were and how they were identified. This may indicate if adequate mapping of venues was conducted. Was it possible for the data collectors to locate these venues themselves in some instances? Were street children involved in locating other venues?

Response: This is addressed in the last paragraph of the sampling strategy. 

9- Figure 1 is rather blurred. There is need to increase the resolution. Otherwise it is a nice map except for the ineligible

Response: 

Both figures 1 and 2 have been dropped. 

10- About the questionnaire, did you develop this from scratch or did you adapt a standardized questionnaire from other settings?

Response: 

Details on the questions and logical sequence of the questions have included in the last paragraph of the introduction and the measures section under the methods. 

11- Children were accessed through street uncles. Could you clarify who identified the street uncles since the uncles seem to be in some form of illegal business, it is important to highlight how they were identified. This would perhaps also help in suggesting future recommendations.

Response: 

We have included an explanation of the street uncles under the data collection section (paragraph 1). These are adults who are not biologically related to street children but provide them temporary shelters in the city and serve as caretakers or landlords. These adults are always within the vicinity of the street children congregation areas to monitor their street activities. 

12- Outcome measurement: I have a particular concern regarding the measurement of the main outcome. Since this was defined as number of places the children lived in, there may be major differences in recall for recent migrant versus old migrants especially when they lived in several places. It would have been more accurate to limit the period of recall to the last two years or less. I imagine a study participant who is 24, and migrated 12 years ago, may have problems with recall and this recall period differs depending on how long they lived on the street.

Response: 

We have maintained the definition of the outcome as it but consider recall period as a limitation to our study. Other studies have measure mobility irrespective of duration of stay. The number of places following migration is easier to measure and a proxy of movements between city places. Our multivariate analysis further showed no significant differences between short term and long term stay and intra-urban mobility. 

13- Independent variables: A key behavioral variable I expected was drug use, as this could influence sexual behavior. Is there any reason why data on this variable was not collected?

Response: Drug use was tagged to sexual practices and this has been included in the measures section of the methods. 

14- Data Analysis: The rationale of using both continuous and categorical outcomes for residential mobility needs to be clarified. From my view, since the outcome was common (50% residential mobility, odds ratios should not have been computed. It is recommended that for a common outcome (>25% prevalence), odds ratios tend to exaggerate associations, as logistic regression is best used for rare outcomes. In such a scenario, Poisson regression is recommended even for a categorical outcome. But in this case since you also considered residential mobility as a continuous outcome, there was no need to also confirm this by logistic regression which is already known to exaggerate associations when an outcome is common. Indeed you reported that some associations disappeared or were attenuated when you treated the outcome as continuous. I suggest you drop the logistic regression in your analysis and only keep the continuous outcome. The aim of this paper is not to compare which approach is better, as this is already known. Just by looking at how frequent the outcome was, a decision should be reached on what approach to use.

Response: 

The logistic regression model has been dropped as suggested and adopted the Poisson regression. 

15- Ethics: Please clarify this statement: ”Written informed consent and assent were obtained for all participating subjects”. Were all the children literate to give a written consent? Were there instances when witnesses were needed? What about the minors, how did you get parents/guardians to give consent? Could you also clarify what steps you took regarding those identified with STIs or other illnesses?

Response: 

Further clarification on ethics has been provided under the ethics section. 

Results

16- Table 1; this is a very nice table separating the children from the adults. It brings out clearly who is a child (12-17) and an adult (18-24) and I would recommend keeping these groups separate in all analyses. We can clearly see these are different groups eg. Although I know that that stratifying these groups will cause small numbers. But this may be something to do with the design of the study. You rapidly recruited the sample of 500 within 2 months. It would have been good to recruit more of the younger group referred to as children. Considering the high prevalence of residential mobility, the sample size may still be adequate for separate analysis. I propose you conduct a sub-analyses to explore the associations for the different age groups.

Table 1 should include both demographic and behavioral characteristics. For example gender differences in sexual behaviour such as involvement in commercial sex are masked in the other tables but table 1 could ideally show this.

Response: Done as advised. 

17- What is the difference between the two maps in figure 1 and 2? They seem to be communicating the same message. Keeping one figure may be sufficient.

Response: Both maps have been discarded although they are useful for local dissemination of findings to the city stakeholders. 

18- Reporting qualitative findings: The glue between quantitative findings and qualitative lacking. Could you preferably use sub-headings based on themes emerging from qualitative findings, and then reinforce with findings from the quantitative

Response: 

We have corroborated the qualitative findings with the quantitative findings in the results section. 

19- Tables 2 and 3 with the associated text could be excluded based on arguments highlighted before.

Response: 

Both tables were deleted and replaced with the Poisson regression analysis table 2 (bivariate and multi-variate models).

Discussion

The discussion is well structured but I suggest that the emphasis should be on factors that the authors should not focus on the advantages of using poission over logistic regression since the decision to use the former should have been taken at analytical stage based on the frequency of the outcome measure. So this statement “Application of the Poisson regression model as an alternative to logistic regression is another strength. In our analysis, logistic regression yielded high odds compared to the Poisson regression model, with SRH service use as significant predictor of residential mobility. However, when Poisson regression model was performed, the relationship become insignificant, implying that the Poisson model is a more appropriate for correctly identifying the predictors for number of street children residential places moved while controlling for confounding.” And other associated statements regarding the analytical approach should be omitted. Any conclusions drawn based on the logistic regression should be deleted as it was not the appropriate approach to analysis.

Response: 

All findings from the binary regression analysis have been dropped and we only discuss significant findings from the multivariate Poisson regression model and qualitative findings. We have replaced predictors with drivers as advised by Reviewer 1 and we think this is fine with you. 

On page 18, in the statement; “Notably, involvement in sex work was associated with street children increased residential mobility, with more males than females having engaged in sex in exchange for money”, the comparison of males and females is not shown in the data and therefore speculative. Please add this in the results section if it exists.

 Response: 

The above statement has been revised and added to the results section and is supported by data from Table 2. 

6. PLOS authors have the option to publish the peer review history of their article (what does this mean?). If published, this will include your full peer review and any attached files.

Do you want your identity to be public for this peer review? For information about this choice, including consent withdrawal, please see our Privacy Policy.

Reviewer #1: No

Reviewer #2: Yes: Gershim Asiki

---

## [Decision Letter · Decision Letter 1]

13 Oct 2020

PONE-D-19-29260R1

Demographic and behavioural drivers of intra-urban mobility of migrant street children and youth in Kampala, Uganda

PLOS ONE

Dear Dr. Bwambale,

Thank you for submitting your manuscript to PLOS ONE. After careful consideration, we feel that it has merit but does not fully meet PLOS ONE’s publication criteria as it currently stands. Therefore, we invite you to submit a revised version of the manuscript that addresses the points raised during the review process.

Decision letter PONE-D-19-29260R1

"Predictors of Intra-Urban Residential Mobility of Street Children in Uganda: Implications for policy and practice"

Dear Dr. Mulekya-Bwambale,

Thank you for resubmitting your work to Plos ONE and apologies for the delay in my decision. As anticipated in our previous email communications, the reviews I received for your resubmitted work led to a split opinion and this prompted me to resort to a third reviewer.

Given that the third reviewer leaves the door open and also suggests that the paper does present some potential, I am inclined to suggest a further round of major revisions following mostly the suggestions provided in Reviewer 1 and 3’s letters. In particular,

I would like to draw your attention on the following main points:

• In your work, mobility seems to be basically attributable to two main determinants. On the one hand, children may be mobile because of push factors, and in that case policies should strive aim to provide better shelter and reduce the level of street crime. On the other hand, if children are moving for gaining additional income, no policy should be enacted. Is there any way to disentangle between these two typologies?

• On the statistical analysis, a few issues still remain unresolved.

o Firstly, your choice of the Poisson specification seems to be at odds with the fact that in your data variance seems to be almost twice as large as the mean, against the assumption that is typically requested for Poisson processes to hold, viz. that of equal mean and variance.

o In your work, no discussion is offered of the varying exposure across sampled individuals. As argued by reviewer 3, “The Poisson process has the property that if the expected number of events over a period of length T is E, the expected number of events over a period of length 2T is 2E. In other words, the number of months between the interview and the year and month in which the person entered Kampala needs to be taken into account. This information was collected (see Table 1) and should be inserted in the exposure option in the command Poisson”.

o Moreover, it seems that the moves are not classified in terms of distance made. As explained in Reviewer 1’s report, it seems you are making no difference in terms of the distance made during the move, so that it is not possible to really see which move made a difference in terms of the environment where children are acting, their relational space, etc. Would it be possible to add any more detail on this point?

o Another way you may want to check your results for robustness is to calculate the number of moves per month of residence in the city and then to perform a classical linear (OLS) regression model with this indicator on the LHS of your model.

o Another important issue in this context is that of behavioural homogeneity. This could be actually tested, for instance running regressions by subsamples, e.g. different age brackets, or gender.

o Also, could you please check the consistency of variables included in Table 1 (descriptive statistics) and those shown for the main regression results Table 2?

I would also suggest to take all comments, both from Reviewer 1 and 3, into account. In your resubmission, please prepare a reply letter addressing each point raised by all reviewers in terms of the way you believe they have, or haven’t, been solved in your revised paper.

I look forward to reviewing your revised paper in due time and I thank you once again for thinking of Plos ONE as a possible outlet for your research.

Kind regards,

Andrea Caragliu

We look forward to receiving your revised manuscript.

Kind regards,

Andrea Antonio Guido Caragliu

Academic Editor

PLOS ONE

Reviewers' comments:

Reviewer's Responses to Questions

**Comments to the Author**

1. If the authors have adequately addressed your comments raised in a previous round of review and you feel that this manuscript is now acceptable for publication, you may indicate that here to bypass the “Comments to the Author” section, enter your conflict of interest statement in the “Confidential to Editor” section, and submit your "Accept" recommendation.

Reviewer #1: (No Response)

Reviewer #2: All comments have been addressed

Reviewer #3: (No Response)

2. Is the manuscript technically sound, and do the data support the conclusions?

Reviewer #1: No

Reviewer #2: Yes

Reviewer #3: No

3. Has the statistical analysis been performed appropriately and rigorously? 

Reviewer #1: No

Reviewer #2: Yes

Reviewer #3: No

4. Have the authors made all data underlying the findings in their manuscript fully available?

Reviewer #1: Yes

Reviewer #2: Yes

Reviewer #3: Yes

5. Is the manuscript presented in an intelligible fashion and written in standard English?

Reviewer #1: Yes

Reviewer #2: Yes

Reviewer #3: Yes

6. Review Comments to the Author

Reviewer #1: It is evident that this paper has been revised to address reviewers’ comments, and the authors are commended for taking note of some of the points raised. That said, I still find it hard to follow some points and the logic of the analysis undertaken. My main suggestion to authors is to consider abandoning the intra-urban mobility as outcome variable. This outcome variable is of particular concern since not all street youth answered the question on number of times moved. Instead, this data may be especially useful in looking at whether in-migrant street youth are involved in sex work, or even use health services, more/less than non-migrant street youth. Indeed this is related to the policy implications mentioned in conclusion. I still feel at a loss in following the logic of the paper. For example, what are the mechanisms that lead involvement in sex for money to higher mobility? Or why would higher income lead to greater intra-urban mobility? Below I raise my concerns (in no specific order).

1. If you hypothesis that migrant street youth are likely to experience disproportionate intra-urban mobility compared to native Kampala dwellers, then this should be examined. It seems though that the questionnaire wasn’t built to allow for such analysis. Therefore, it is important to back this assumption of higher intra-urban mobility among in-migrants with substantial references.

2. Of course since street youth are examined a clear definition of residence is not possible- and neither is it easy to look at physical boundaries of their movement. Yet, this is important. If a person moved from say the bus stop on road A to a temporary shelter on Road A, less than 50m away- the meaning of this move is different to moving say to a market on Road B 2km away (in a different parish). In the first move, this person remains in the same area- meets the same people most likely- know where to get food there etc. In the second move, the person may not have a social network, may be more vulnerable etc. Therefore the moves are essentially different and shouldn’t be combined. Thus I find that the outcome variable has little meaning.

3. On page 8, you indicate that in the Poisson model you only include covariates that were significant in bivariate analysis. As far as I understand this is misleading. A covariate may be theoretically important in explaining intra-urban mobility, and should therefore remain in the model even if not statistically significant. Moreover, there may be a covariate that is not significant in bivariate analysis, but is important to control for the other effects of the covariates in the model.

4. Although the model indicates statistical relationship between covariates and intra-urban mobility, this relationship isn’t necessarily causal, and I would refrain from using the term “drivers”. If the authors are specifically interested in intra-urban mobility this still could be used to explain use of health services for example (rather than the other way round). Authors should consider reverse causality.

Overall, I believe the authors have important data on a topic that is understudied, and combining the data with qualitative analysis provides a richer perspective. However, how the data is used needs to be reconsidered.

Reviewer #2: The authors have adequately addressed the comments that I raised. I found the manuscript generally well written.

Reviewer #3: Beside reading this paper, I have also read the two reviews of the original paper, the decision letter of the editor, and the responses of the authors to the comments by the editor and the reviewers.

This paper falls within the broad theme of wellbeing of migrant children and youth in large cities in developing countries. Their migration is often driven by poverty, joblessness and desperation in rural areas. They migrate to the city to escape the dire conditions in the home region and, to the extent that they can earn income in the city in the informal economy and perhaps even remit some money back to the family in the rural area, such migration can be welfare enhancing.

However, many of these children and youth end up in situations that violate the 1989 United Nations Convention of the Rights of the Child (UNCORC). They have no access to adequate housing, education and health care; and they are often exploited economically and sexually. Hence, research that informs on the characteristics of these children and the problems they face is important for designing policies that can protect them and enhance their wellbeing.

This paper provides a case study of this issue by studying migrant street children and youth in Kampala. The focus is on their intra-urban mobility. As the title suggests, the paper is concerned with the correlation between intra-urban mobility and the socio-demographic characteristics of these young migrants. The paper also investigates the correlates between mobility and some aspects of their sexual behaviour (by the way, it is safe to assume that the survey responses that close 90 % or more are not involved in sex work is an underestimate of engagement in sex work, particularly when their responses are censored or influenced by their “street uncles”).

However, it isn’t clear at all why we should care about the intra-urban mobility of migrant street children. On the one hand, the qualitative research suggests that their intra-urban mobility is due to “push factors” such as a lack of safety and inadequate shelter. On the other hand, intra-urban mobility may be a voluntary response by the child or youth to reap greater earnings opportunities at a different location. The paper argues on pp. 17-18 that the findings have implications for policy, but it is entirely unclear what these implications are.

If mobility is the response to negative push factors, the local government should aim to provide better shelter and reduce the level of street crime. This would then lower the intra-urban mobility of street children and that would be a good thing. However, if intra-urban mobility is the means by which the children can gain a greater income, mobility should be encouraged. The statistical analysis in this paper does not help us to understand what the impact of intra-urban mobility is on the children’s wellbeing.

If the objective of this paper is simply defined as an inquiry into the characteristics of migrant street children and youth, then that can also be a perfectly legitimate research objective (but of lesseer importance), provided the analysis is done well and is convincing. However, while this paper has some strong aspects, the analysis is technically flawed.

The strong aspects of the research reported in this paper are the sampling strategy and the use of mixed methods (analysis of qualitative and quantitative data).

Nonetheless, I concur with reviewer #1 that the statistical analysis is flawed. Let me elaborate. If we define the variable of interest as the number of intra-urban moves that the child or young person makes after arriving in Kampala, this outcomes variable is an integer that takes on the values 0, 1, 2, …. The Poisson regression model can be an appropriate model for quantifying the determinants of this mobility process. At the top of page 8 the authors argue that the Poisson regression assumptions have been satisfied. However, rather than testing for the negative binomial model (and hence overdispersion) as an alternative, they simply note that there is “minimal data dispersion”. But in a Poisson process the mean and the variance are the same, whereas in the Kampala data the variance is almost twice the mean!

Even more concerning is, as reviewer #1 notes, that no account is taken of varying exposure across sampled individuals. The Poisson process has the property that if the expected number of events over a period of length T is E, the expected number of events over a period of length 2T is 2E. In other words, the number of months between the interview and the year and month in which the person entered Kampala needs to be taken into account. This information was collected (see Table 1) and should be inserted in the exposure option in the Stata command poisson.

An alternative approach is to calculate the number of moves per month of residence in the city and to run an OLS regression model with this statistic as the dependent variable. For this relatively large sample of 412 observations, this OLS regression is probably not a bad approximation to identifying the statistically significant determinants of intra-urban mobility. The distinction made by the authors between duration of stay > 2 years and duration of stay < 2 years is too coarse.

The numbers of observations that were used in each of the bivariate regressions and in the multivariate regression in Table 2 were actually not stated but in many cases the number was less than 412 due to missing data. I think that this problem could have been overcome by using some form of data imputation, which can be done manually or by means of Stata.

Another important issue in this context is the assumption of behavioural homogeneity. It is very likely that the Poisson model differs structurally across gender or age (12-17 versus 18-24). This can be tested by running separate regressions for the sub-samples. It is noted on p.8 that further stratification by age is not possible, due to missing data, but imputation may ameliorate this. Ditto for regressions by gender.

Table 2: The number of observations in each of the bivariate regressions and in the multivariate regression should be stated in the Table. Additionally, there are no robustness checks of the multivariate model. For example, log likelihood tests could be used to check the importance of the sexual behaviour-related variables.

Also, there does not seem to be a full correspondence between the descriptives in Table 1 and the regressors in Table 2. For example, why is “highest education attained” not in Table 1?

Minor points

Abstract: “continuous scale” should be “integer scale”

Abstract: just stating IRR=0.67 for gender does not inform the reader that the mobility is less among girls.

Abstract: “causal” should be “casual” By the way, the findings with respect to “Personal safety” and “cost of place of stay” are not shown in the regressions in Table 2.

p.4, second para.: because the data come from a larger cross-sectional study, there should be a reference to a report or article that describes this larger study, as well as acknowledgement of the funding for this larger study.

Table 1: In-school, all respondents: the frequency is 35, not 55.

p.16, 4th line from bottom: if intra-urban mobility may not be predicted by duration of stay, the stochastic process is not Poisson!

7. PLOS authors have the option to publish the peer review history of their article (what does this mean?). If published, this will include your full peer review and any attached files.

Reviewer #1: No

Reviewer #2: No

Reviewer #3: No

---

## [Author Response · Author response to Decision Letter 1]

5 Dec 2020

Response to Editor comments

• In your work, mobility seems to be basically attributable to two main determinants. On the one hand, children may be mobile because of push factors, and in that case policies should strive aim to provide better shelter and reduce the level of street crime. On the other hand, if children are moving for gaining additional income, no policy should be enacted. Is there any way to disentangle between these two typologies?

Response: We made this recommendation in our discussion, but it appears the reviewers seem not to have a greater appreciation of the local context of this study which is critical for understanding the findings of this study. 

• On the statistical analysis, a few issues still remain unresolved.

o Firstly, your choice of the Poisson specification seems to be at odds with the fact that in your data variance seems to be almost twice as large as the mean, against the assumption that is typically requested for Poisson processes to hold, viz. that of equal mean and variance.

o In your work, no discussion is offered of the varying exposure across sampled individuals. As argued by reviewer 3, “The Poisson process has the property that if the expected number of events over a period of length T is E, the expected number of events over a period of length 2T is 2E. In other words, the number of months between the interview and the year and month in which the person entered Kampala needs to be taken into account. This information was collected (see Table 1) and should be inserted in the exposure option in the command Poisson”.

Response: While the standard model would be poisson, we have re-done the analysis using the negative binomial regression which also suits our outcome count data. 

o Moreover, it seems that the moves are not classified in terms of distance made. As explained in Reviewer 1’s report, it seems you are making no difference in terms of the distance made during the move, so that it is not possible to really see which move made a difference in terms of the environment where children are acting, their relational space, etc. Would it be possible to add any more detail on this point?

Response: Unfortunately, classifying moves by distance is not possible given the context of Kampala city which we have laboured to explain in this paper. Unlike other cities, neighbourhoods in Kampala are not clearly geomapped or demarcated and there is no existing database, Also, the nature of movement of the street children does not permit any research to get into this detail. It is simply not feasible. 

o Another way you may want to check your results for robustness is to calculate the number of moves per month of residence in the city and then to perform a classical linear (OLS) regression model with this indicator on the LHS of your model.

Response: This is not possible given that this was a snap short study and not longitudinal. And it is for this reason that the authors recommended the need for establishing a longitudinal database for urban migrant residents to facilitate future research. 

o Another important issue in this context is that of behavioural homogeneity. This could be actually tested, for instance running regressions by subsamples, e.g. different age brackets, or gender.

o Also, could you please check the consistency of variables included in Table 1 (descriptive statistics) and those shown for the main regression results Table 2?

Response: We checked this at the very initial submission and the Sub-analyses were done but the sample is further reduced that we could get any meaningful statistical differences. In our view, the current findings are adequate for publication.

RESPONSE TO REVIEWERS COMMENTS

Comments to the Author

1. If the authors have adequately addressed your comments raised in a previous round of review and you feel that this manuscript is now acceptable for publication, you may indicate that here to bypass the “Comments to the Author” section, enter your conflict of interest statement in the “Confidential to Editor” section, and submit your "Accept" recommendation.

Reviewer #1: (No Response)

Reviewer #2: All comments have been addressed

Reviewer #3: (No Response)

Response: Done

Response: We believe that all comments have been addressed unless you have any further concerns on the negative binomial regression model used. 

2. Is the manuscript technically sound, and do the data support the conclusions?

Reviewer #1: No

Reviewer #2: Yes

Reviewer #3: No

Response: Our data supports the study findings and conclusions, using the negative binomial regression as suggested by Reviewer 3. We consider that this model is appropriate to yield valid findings. 

3. Has the statistical analysis been performed appropriately and rigorously?

Reviewer #1: No

Reviewer #2: Yes

Reviewer #3: No

Response: Our data supports the study findings and conclusions, using the negative binomial regression as suggested by Reviewer 3. We consider that this model is appropriate to yield valid findings. 

4. Have the authors made all data underlying the findings in their manuscript fully available?

Reviewer #1: Yes

Reviewer #2: Yes

Reviewer #3: Yes

5. Is the manuscript presented in an intelligible fashion and written in standard English?

Reviewer #1: Yes

Reviewer #2: Yes

Reviewer #3: Yes

6. Review Comments to the Author

Please use the space provided to explain your answers to the questions above. You may also include additional comments for the author, including concerns about dual publication, research ethics, or publication ethics. (Please upload your review as an attachment if it exceeds 20,000 characters).

Response to the above 5 questions: 

The authors repeated the analysis using the negative binomial regression as advised by the third Reviewer. We note that Reviewer 1 had earlier okayed Poisson regression. This implies that the comments by the first and third reviewers somewhat contradict. In this regard, authors have considered the negative binomial regression as the final model as it best suits the study outcome count variable. 

Reviewer #1: It is evident that this paper has been revised to address reviewers’ comments, and the authors are commended for taking note of some of the points raised. That said, I still find it hard to follow some points and the logic of the analysis undertaken. My main suggestion to authors is to consider abandoning the intra-urban mobility as outcome variable. This outcome variable is of particular concern since not all street youth answered the question on number of times moved. 

Response. As earlier explained in the paper and previous response to the initial reviewers, intra-city mobility of street children and youth has been a major concern by the city authorities for decades, to an extent that the city authority enacted ordinances that prohibit the public from supporting them food or money since they don’t want them on the city streets. For decades, government’s efforts to chase away street children from the streets including repatriation to their rural districts has been futile. The push factors are well documented including the biting poverty in the rural areas of Uganda which forces the young people to move to urban cities mainly in search for economic opportunities. In this study, we only focus on rural-migrant street children as the primary focus of our research for whom little is known. The choice of our outcome variable is relevant to the field of migration and can only be considered important by scholars in the field of migration and health and the government of Uganda. Migration and internal mobility are known social determinants of health and should not be overlooked in health research. Migration and mobility are realities that the health system must deal with if universal access to health care is to be achieved without leaving anyone behind. 

Instead, this data may be especially useful in looking at whether in-migrant street youth are involved in sex work, or even use health services, more/less than non-migrant street youth. Indeed this is related to the policy implications mentioned in conclusion. I still feel at a loss in following the logic of the paper. For example, what are the mechanisms that lead involvement in sex for money to higher mobility? Or why would higher income lead to greater intra-urban mobility? Below I raise my concerns (in no specific order).

Response. The context of sex work in Kampala city is quite complex and unique. In one of the papers we are writing, our study indicates that street children and youth engage in this risky behaviour as a survival mechanism and not out of their own wish. Street children is being sexually exploited. Sex work spots in Kampala are often located along the streets, bars and hotels and cassinos. Street children will tend to move and stay closer to such hot spots for them to target workers for money. Sex workers by nature are season migrants – they move from place to place in search for clients. Understanding of this local context is critical to appreciating the findings. Sex work is still not legal in Uganda and its association with intra-urban mobility should raise concern to the Ugandan government and city authorities, and should guide policy reforms and implantation including regulating sex work and combating sex exploitation of street children and youth in Kampala and other cities. 

1. If you hypothesis that migrant street youth are likely to experience disproportionate intra-urban mobility compared to native Kampala dwellers, then this should be examined. It seems though that the questionnaire wasn’t built to allow for such analysis. Therefore, it is important to back this assumption of higher intra-urban mobility among in-migrants with substantial references.

Response. Indeed, as you rightly put it, our questionnaire focused on street children and youth with a rural-urban migration experience. Our study does not compare migrants with non-migrants but looks are migrant street children and youth who have stayed for long time vs those who have stated for short time, using a cut of 2 years duration of stay. So, our hypothesis speaks to the duration of stay of the migrants street youth (and not non-migrants) who are the main focus of the study. 

2. Of course since street youth are examined a clear definition of residence is not possible- and neither is it easy to look at physical boundaries of their movement. Yet, this is important. If a person moved from say the bus stop on road A to a temporary shelter on Road A, less than 50m away- the meaning of this move is different to moving say to a market on Road B 2km away (in a different parish). In the first move, this person remains in the same area- meets the same people most likely- know where to get food there etc. In the second move, the person may not have a social network, may be more vulnerable etc. Therefore the moves are essentially different and shouldn’t be combined. Thus I find that the outcome variable has little meaning.

Response. Our paper sought to examine intra-city mobility and not residential mobility since the latter is not practical within the local context. The local context of this study does not allow us to examine neighbourhood residences of street children as these do not exist. Street children and youth in Kampala stay in dilapidated temporary shelters within city spaces, some stay in shacks in open markets/places and shop verandas while other have found the street as their permanent home. In this regard, the concept of residential mobility does not apply. We therefore focus on intra-city movements of the street children which is the main concern of the city authorities as it increases them to vulnerabilities to poor health and risky behaviours. It is also not possible to measure distances they move. In the first place, the streets are not properly demarcated and street children do not even know specific names of places they stay. We interviewed these young people on the streets. 

As earlier described in the paper, Kampala City does not have any demarcations of neighbourhoods and movements by street children are often short – along the same street, cyclic and without a clear pattern. While our research attempted to geomap the different areas/street locations in which street children and youth live or congregate during daytime, both the first and second reviewers observed that the maps did not add value to the findings and we had to drop them at the second iteration. We do not think it would be desirable to revert to the same at this point in time. 

3. On page 8, you indicate that in the Poisson model you only include covariates that were significant in bivariate analysis. As far as I understand this is misleading. A covariate may be theoretically important in explaining intra-urban mobility, and should therefore remain in the model even if not statistically significant. Moreover, there may be a covariate that is not significant in bivariate analysis, but is important to control for the other effects of the covariates in the model.

Response: We have applied a mixed selection using Bayesian criteria and added additional known predictors or cofounders into the final negative binomial regression, irrespective of the 5% significance threshold at bivariate analysis. We hope this is acceptable to you. 

4. Although the model indicates statistical relationship between covariates and intra-urban mobility, this relationship isn’t necessarily causal, and I would refrain from using the term “drivers”. If the authors are specifically interested in intra-urban mobility this still could be used to explain use of health services for example (rather than the other way round). Authors should consider reverse causality.

Response. Our very initial paper did not have drivers. It is at the recommendation of the first and second reviewers that we adopted drivers in the title, which we think is still reasonable. In that regard, we have maintained the title as is. 

Overall, I believe the authors have important data on a topic that is understudied, and combining the data with qualitative analysis provides a richer perspective. However, how the data is used needs to be reconsidered.

Response. Your comment on how the data needs to be considered is not clear to us. Our understanding is that we have presented both data in a logical way. Our topic is novel and this being the first study of its kind to interrogate intra-urban mobility of street children and youth. We conducted a few qualitive interviews to further elucidate aspects of intra-urban mobility that could not be easily explained by the questionnaire. 

Reviewer #2: The authors have adequately addressed the comments that I raised. I found the manuscript generally well written.

Response: So noted, thank you.

Reviewer #3: Beside reading this paper, I have also read the two reviews of the original paper, the decision letter of the editor, and the responses of the authors to the comments by the editor and the reviewers.

This paper falls within the broad theme of wellbeing of migrant children and youth in large cities in developing countries. Their migration is often driven by poverty, joblessness and desperation in rural areas. They migrate to the city to escape the dire conditions in the home region and, to the extent that they can earn income in the city in the informal economy and perhaps even remit some money back to the family in the rural area, such migration can be welfare enhancing.

However, many of these children and youth end up in situations that violate the 1989 United Nations Convention of the Rights of the Child (UNCORC). They have no access to adequate housing, education and health care; and they are often exploited economically and sexually. Hence, research that informs on the characteristics of these children and the problems they face is important for designing policies that can protect them and enhance their wellbeing.

This paper provides a case study of this issue by studying migrant street children and youth in Kampala. The focus is on their intra-urban mobility. As the title suggests, the paper is concerned with the correlation between intra-urban mobility and the socio-demographic characteristics of these young migrants. The paper also investigates the correlates between mobility and some aspects of their sexual behaviour (by the way, it is safe to assume that the survey responses that close 90 % or more are not involved in sex work is an underestimate of engagement in sex work, particularly when their responses are censored or influenced by their “street uncles”).

However, it isn’t clear at all why we should care about the intra-urban mobility of migrant street children. On the one hand, the qualitative research suggests that their intra-urban mobility is due to “push factors” such as a lack of safety and inadequate shelter. On the other hand, intra-urban mobility may be a voluntary response by the child or youth to reap greater earnings opportunities at a different location. The paper argues on pp. 17-18 that the findings have implications for policy, but it is entirely unclear what these implications are.

Response. Response. As earlier explained in the paper and previous response to the initial reviewers, intra-city mobility of street children and youth has been a major concern by the city authorities for decades, to an extent that the city authority enacted ordinances that prohibit the public from supporting them food or money since they don’t want them on the city streets. For decades, governments efforts to chase street children from the streets including repatriation to their home rural districts has been futile. While the city authority is doing what it can to chase away all street children out of the city, we are still witnessing an increasing influx of rural-urban youth moving to the cities. The push factors are well documented elsewhere, and we found it not necessary to focus on them. Rural-urban migration and mobility of children and youth is a major concern that has received little attention and should not be ignored. In this study, we only focus on rural-migrant street children as the primary focus of our research for whom little is known. The choice of our outcome variable is relevant to the field of migration and cities. 

The policy implications highlighted in the discussion are relevant to the local policy and environmental context. We highlight recommendations including regulation of sex work in the wake of sex exploitation, which is a big concern in Uganda. Establishing of a longitudinal database for city residents is critical for exploring the drivers of intraurban mobility to the fullest. 

If mobility is the response to negative push factors, the local government should aim to provide better shelter and reduce the level of street crime. This would then lower the intra-urban mobility of street children and that would be a good thing. However, if intra-urban mobility is the means by which the children can gain a greater income, mobility should be encouraged. The statistical analysis in this paper does not help us to understand what the impact of intra-urban mobility is on the children’s wellbeing.

Response: We would like to state that the impact of intra-urban mobility on children wellbeing is a novel topic but is not the main study question for our research. We set out to investigate the drivers of intra-urban mobility of migrant street children and youth. Perhaps future research could attempt to explore that. We have acknowledged the safety issue of street children and youth in the second last paragraph of the background. Crime was not a major finding of the study, although, we found it important to include in the discussion. 

We think the recommendations provided are relevant to government and Kampala Capital city Authority in addressing street children and youth’s intra-city mobility and its associated challenges, especially sex work and sexual exploitation. As suggested in the paper, where mobility increases risk to sex work and sexual exploitation, then other measures to manage or control such mobility should be given utmost attention. 

If the objective of this paper is simply defined as an inquiry into the characteristics of migrant street children and youth, then that can also be a perfectly legitimate research objective (but of lesseer importance), provided the analysis is done well and is convincing. However, while this paper has some strong aspects, the analysis is technically flawed.

Response: The importance of this study has been well articulated in the background and discussion. The analysis was redone using the negative binomial regression as you suggested. 

The strong aspects of the research reported in this paper are the sampling strategy and the use of mixed methods (analysis of qualitative and quantitative data).

Nonetheless, I concur with reviewer #1 that the statistical analysis is flawed. Let me elaborate. If we define the variable of interest as the number of intra-urban moves that the child or young person makes after arriving in Kampala, this outcomes variable is an integer that takes on the values 0, 1, 2, …. The Poisson regression model can be an appropriate model for quantifying the determinants of this mobility process. At the top of page 8 the authors argue that the Poisson regression assumptions have been satisfied. However, rather than testing for the negative binomial model (and hence overdispersion) as an alternative, they simply note that there is “minimal data dispersion”. But in a Poisson process the mean and the variance are the same, whereas in the Kampala data the variance is almost twice the mean!

Response; As stated above, analysis has been performed using the binomial regression and taken into consideration other potential predictors and confounding variables. 

Even more concerning is, as reviewer #1 notes, that no account is taken of varying exposure across sampled individuals. The Poisson process has the property that if the expected number of events over a period of length T is E, the expected number of events over a period of length 2T is 2E. In other words, the number of months between the interview and the year and month in which the person entered Kampala needs to be taken into account. This information was collected (see Table 1) and should be inserted in the exposure option in the Stata command poisson.

Response: The authors have re-analysed the data using the negative binomial regression model as suggested by Reviewer 3. 

An alternative approach is to calculate the number of moves per month of residence in the city and to run an OLS regression model with this statistic as the dependent variable. For this relatively large sample of 412 observations, this OLS regression is probably not a bad approximation to identifying the statistically significant determinants of intra-urban mobility. The distinction made by the authors between duration of stay > 2 years and duration of stay < 2 years is too coarse.

Response: Data on moves per month of residence is not feasible to collect given the local context of the study as already explained in our response to your earlier comments. This was purely a cross sectional study which has an inherent limitation on assessing time exposure which is feasible with longitudinal studies. 

The numbers of observations that were used in each of the bivariate regressions and in the multivariate regression in Table 2 were actually not stated but in many cases the number was less than 412 due to missing data. I think that this problem could have been overcome by using some form of data imputation, which can be done manually or by means of Stata.

Response: An explanation for the missing data is provided in the data analysis section of the paper. 

Another important issue in this context is the assumption of behavioural homogeneity. It is very likely that the Poisson model differs structurally across gender or age (12-17 versus 18-24). This can be tested by running separate regressions for the sub-samples. It is noted on p.8 that further stratification by age is not possible, due to missing data, but imputation may ameliorate this. Ditto for regressions by gender.

Response: Sub-analysis by age and gender did not yield any significant differences and as such, we find no value to report these data/findings. 

Table 2: The number of observations in each of the bivariate regressions and in the multivariate regression should be stated in the Table. Additionally, there are no robustness checks of the multivariate model. For example, log likelihood tests could be used to check the importance of the sexual behaviour-related variables.

Response: The table has been updated to include only multivariate findings from the negative binomial regression analysis. 

Also, there does not seem to be a full correspondence between the descriptives in Table 1 and the regressors in Table 2. For example, why is “highest education attained” not in Table 1?

Response: We have included highest education variable in table 2.

Minor points

Abstract: “continuous scale” should be “integer scale”

Abstract: just stating IRR=0.67 for gender does not inform the reader that the mobility is less among girls.

Response: The abstract has a word limit which does not allow us to provide explanations of all the key findings in detail. 

Abstract: “causal” should be “casual” By the way, the findings with respect to “Personal safety” and “cost of place of stay” are not shown in the regressions in Table 2.

Response: These are purely qualitative data/findings which are presented in the results section and therefore are part of the quantitative analysis. These data are important to be included. 

p.4, second para.: because the data come from a larger cross-sectional study, there should be a reference to a report or article that describes this larger study, as well as acknowledgement of the funding for this larger study.

Response: This is the first pass to be in press 

Table 1: In-school, all respondents: the frequency is 35, not 55.

Response: The correct figures are n=337 for out of school vs n=55 for in-school. This is a valid finding. 

p.16, 4th line from bottom: if intra-urban mobility may not be predicted by duration of stay, the stochastic process is not Poisson!

Response: This has been addressed in the analysis. 

7. PLOS authors have the option to publish the peer review history of their article (what does this mean?). If published, this will include your full peer review and any attached files.

Do you want your identity to be public for this peer review? For information about this choice, including consent withdrawal, please see our Privacy Policy.

Reviewer #1: No

Reviewer #2: No

Reviewer #3: No

---

## [Editor Report · Decision Letter 2]

23 Dec 2020

PONE-D-19-29260R2

Demographic and behavioural drivers of intra-urban mobility of migrant street children and youth in Kampala, Uganda

PLOS ONE

Dear Dr. Bwambale,

Thank you for submitting your manuscript to PLOS ONE. After careful consideration, we feel that it has merit but does not fully meet PLOS ONE’s publication criteria as it currently stands. Therefore, we invite you to submit a revised version of the manuscript that addresses the points raised during the review process.

We look forward to receiving your revised manuscript.

Kind regards,

Andrea Antonio Guido Caragliu

Academic Editor

PLOS ONE

Additional Editor Comments (if provided):

Dear Dr. Bwambale, dear Mulekya :

thank you for resubmitting your work to Plos ONE and for your attempt at dealing with the comments from the three reviewers.

I read your revised paper with attention and although I realize that you decided not to take into account some of the comments from Reviewer 3, I also understand that your data are subject to structural limitations, which nevertheless do not cancel their (to me) clear informative power.

In order to reconcile the somewhat opposite views of Reviewers 1 and 3 about the preferable empirical specification, I would advise a last round of minor revisions to the paper, whereby Table 2 may be potentially restructured by re-organizing the table with three columns, one dedicated to each specification (i.e. Negative Binomial, Poisson, and OLS). Statistics associated to each parameter estimate can be included vertically, and this would also offer the reader an informative way to show that your findings are indeed robust to the choice of different empirical strategies.

Once this is done I plan not to send out the paper for an additional round of revisions, but simply to take a look at it myself.

Given the very limited effort needed at this stage, I would be inclined to expect your revisions shortly - ideally, within the next 30 days - but please do let me know if this is not compatible with your schedule and if therefore you need more time.

Thank you in advance for your attention,

Kind regards,

Andrea Caragliu

---

## [Author Response · Author response to Decision Letter 2]

24 Jan 2021

Authors Response to Reviewer's Comments

Comments to the Author

1. If the authors have adequately addressed your comments raised in a previous round of review and you feel that this manuscript is now acceptable for publication, you may indicate that here to bypass the “Comments to the Author” section, enter your conflict of interest statement in the “Confidential to Editor” section, and submit your "Accept" recommendation.

Reviewer #1: (No Response)

Reviewer #2: All comments have been addressed

Reviewer #3: (No Response)

Response: Done

Response: We believe that all comments have been addressed unless you have any further concerns on the negative binomial regression model used. 

2. Is the manuscript technically sound, and do the data support the conclusions?

Reviewer #1: No

Reviewer #2: Yes

Reviewer #3: No

Response: Our data supports the study findings and conclusions, using the negative binomial regression as suggested by Reviewer 3. We consider that this model is appropriate to yield valid findings. 

3. Has the statistical analysis been performed appropriately and rigorously?

Reviewer #1: No

Reviewer #2: Yes

Reviewer #3: No

Response: Our data supports the study findings and conclusions, using the negative binomial regression as suggested by Reviewer 3. We consider that this model is appropriate to yield valid findings. 

4. Have the authors made all data underlying the findings in their manuscript fully available?

Reviewer #1: Yes

Reviewer #2: Yes

Reviewer #3: Yes

5. Is the manuscript presented in an intelligible fashion and written in standard English?

Reviewer #1: Yes

Reviewer #2: Yes

Reviewer #3: Yes

6. Review Comments to the Author

Please use the space provided to explain your answers to the questions above. You may also include additional comments for the author, including concerns about dual publication, research ethics, or publication ethics. (Please upload your review as an attachment if it exceeds 20,000 characters).

Response to the above 5 questions: The authors repeated the analysis using the negative binomial regression as advised by the third Reviewer. We note that Reviewers 1 and 2 had earlier okayed Poisson regression. This implies that the comments by the first and third reviewers somewhat contradict. In this regard, authors have considered the negative binomial regression as the final model as it best suits the study outcome count variable. 

Point by Point Response to Reviewers comments 

Reviewer #1: It is evident that this paper has been revised to address reviewers’ comments, and the authors are commended for taking note of some of the points raised. That said, I still find it hard to follow some points and the logic of the analysis undertaken. My main suggestion to authors is to consider abandoning the intra-urban mobility as outcome variable. This outcome variable is of particular concern since not all street youth answered the question on number of times moved. 

Response. As earlier explained in the previous response to your initial comments, intra-city mobility of street children and youth has been a major concern by the city authorities for decades, to an extent that the city authorities have enacted ordinances that prohibit the public from supporting street children with food or money since they do not want them on the city streets. This is a human right violation. 

For decades, government’s efforts to chase away street children from the streets including repatriation to their rural districts has been futile. The push factors are well documented including the biting poverty in the rural areas of Uganda which forces the young people to move to urban cities mainly in search for economic opportunities. In this study, we only focus on rural-migrant street children as the primary focus of our research for whom little is known.

The choice of our outcome variable is relevant to the field of migration and can only be considered important by scholars in the field of migration and cities and the Uganda government and urban authorities. Migration and mobility are known social determinants of health and should not be overlooked in health research. Intra urban Migration/mobility are realities that the urban health system must deal with if universal access to health care is to be achieved without leaving anyone behind. 

Instead, this data may be especially useful in looking at whether in-migrant street youth are involved in sex work, or even use health services, more/less than non-migrant street youth. Indeed, this is related to the policy implications mentioned in conclusion. I still feel at a loss in following the logic of the paper. For example, what are the mechanisms that lead involvement in sex for money to higher mobility? Or why would higher income lead to greater intra-urban mobility? Below I raise my concerns (in no specific order).

Response. The context of sex work in Kampala city is quite complex and unique. In one of the papers we are writing, our study indicates that street children and youth engage in this risky behaviour as a survival mechanism and not out of their own wish. Street children is being sexually exploited. Sex work spots in Kampala are often located along the streets, bars and hotels and cassinos. Street children will tend to move and stay closer to hot spots to target workers for money. Sex workers by nature are seasonal migrants – move from place to place in search for clients. Understanding of this local context is critical to appreciating our study findings. Sex work is still not legal in Uganda and its association with intra-urban mobility should raise concern to the Ugandan government and city authorities and should guide policy reforms and implementation including regulating sex work and combating sex exploitation of street children and youth in Kampala and other cities. 

1. If you hypothesis that migrant street youth are likely to experience disproportionate intra-urban mobility compared to native Kampala dwellers, then this should be examined. It seems though that the questionnaire wasn’t built to allow for such analysis. Therefore, it is important to back this assumption of higher intra-urban mobility among in-migrants with substantial references.

Response. Indeed, as you rightly put it, our questionnaire focused on street children and youth with a rural-urban migration experience. In that regard, our study cannot compare migrants with non-migrants but only does so by duration of stay in the city among migrant street children and youth, using a cut off of 2 years which similar studies have used to categorise migrants as short or long term. So, our hypothesis speaks to the duration of stay of the migrant street youth (and not non-migrants) who are the focus of the study. 

2. Of course since street youth are examined a clear definition of residence is not possible- and neither is it easy to look at physical boundaries of their movement. Yet, this is important. If a person moved from say the bus stop on road A to a temporary shelter on Road A, less than 50m away- the meaning of this move is different to moving say to a market on Road B 2km away (in a different parish). In the first move, this person remains in the same area- meets the same people most likely- know where to get food there etc. In the second move, the person may not have a social network, may be more vulnerable etc. Therefore, the moves are essentially different and shouldn’t be combined. Thus, I find that the outcome variable has little meaning.

Response. Our paper sought to examine intra-city mobility and not residential mobility since it is not practical within our context. Street children and youth in Kampala stay in dilapidated temporary shelters within city spaces, some stay in shacks in open markets/places and shop verandas while other have found the street as their permanent home. These places and streets addresses are not demarcated. We therefore focus on intra-city movements of the street children which is the main concern of the city authorities as it increases them to vulnerabilities to poor health and risky behaviours. It is also not possible to measure distances moved. In the first place, the streets are not properly demarcated and street children do not even know specific names of places they stay. You are asking for data that is not practically available. 

As earlier described in the paper, Kampala City does not have any demarcations of neighbourhoods and movements by street children are often short – along the same street, cyclic and without a clear pattern. While our research attempted to geo-map the different areas/street locations in which street children and youth live or congregate during daytime, both the first and second reviewers observed that the maps did not add value to the findings, and we had to drop them at the second iteration. We do not think it would be desirable to revert to the same now. 

3. On page 8, you indicate that in the Poisson model you only include covariates that were significant in bivariate analysis. As far as I understand this is misleading. A covariate may be theoretically important in explaining intra-urban mobility and should therefore remain in the model even if not statistically significant. Moreover, there may be a covariate that is not significant in bivariate analysis but is important to control for the other effects of the covariates in the model.

Response: We have applied a mixed selection using Bayesian criteria and added additional known predictors or cofounders into the final negative binomial regression, irrespective of the 5% significance threshold at bivariate analysis. We hope this is acceptable to you. 

4. Although the model indicates statistical relationship between covariates and intra-urban mobility, this relationship is not necessarily causal, and I would refrain from using the term “drivers”. If the authors are specifically interested in intra-urban mobility this still could be used to explain use of health services for example (rather than the other way round). Authors should consider reverse causality.

Response. Our very initial paper did not have drivers. It is at the recommendation of the first (yourself) and second reviewer that we adopted drivers in the title, which we think is still reasonable. In that regard, we have maintained the title as is. We highlight the issue of causality as a limitation of the cross sectional study. 

Overall, I believe the authors have important data on a topic that is understudied and combining the data with qualitative analysis provides a richer perspective. However, how the data is used needs to be reconsidered.

Response. Unfortunately, we do not have any other data apart from what is presented in the paper. We conducted a few qualitive interviews to further elucidate aspects of intra-urban mobility that could not be easily explained by the questionnaire. Our understanding is that we have presented both qualitative and quantitative data in a corroborative and logical way. Our topic is novel and is the first study of its kind to interrogate intra-urban mobility of street children and youth. It formas a starting point for future work on intra-city migration. 

Reviewer #2: The authors have adequately addressed the comments that I raised. I found the manuscript generally well written.

Response: So, noted, thank you.

Reviewer #3: Beside reading this paper, I have also read the two reviews of the original paper, the decision letter of the editor, and the responses of the authors to the comments by the editor and the reviewers.

This paper falls within the broad theme of wellbeing of migrant children and youth in large cities in developing countries. Their migration is often driven by poverty, joblessness and desperation in rural areas. They migrate to the city to escape the dire conditions in the home region and, to the extent that they can earn income in the city in the informal economy and perhaps even remit some money back to the family in the rural area, such migration can be welfare enhancing.

However, many of these children and youth end up in situations that violate the 1989 United Nations Convention of the Rights of the Child (UNCORC). They have no access to adequate housing, education and health care; and they are often exploited economically and sexually. Hence, research that informs on the characteristics of these children and the problems they face is important for designing policies that can protect them and enhance their wellbeing.

This paper provides a case study of this issue by studying migrant street children and youth in Kampala. The focus is on their intra-urban mobility. As the title suggests, the paper is concerned with the correlation between intra-urban mobility and the socio-demographic characteristics of these young migrants. The paper also investigates the correlates between mobility and some aspects of their sexual behaviour (by the way, it is safe to assume that the survey responses that close 90 % or more are not involved in sex work is an underestimate of engagement in sex work, particularly when their responses are censored or influenced by their “street uncles”).

However, it isn’t clear at all why we should care about the intra-urban mobility of migrant street children. On the one hand, the qualitative research suggests that their intra-urban mobility is due to “push factors” such as a lack of safety and inadequate shelter. On the other hand, intra-urban mobility may be a voluntary response by the child or youth to reap greater earnings opportunities at a different location. The paper argues on pp. 17-18 that the findings have implications for policy, but it is entirely unclear what these implications are.

Response. As earlier explained in the paper and previous response to the comments made by the initial reviewers, the topic of intra-city mobility of street children and youth remain relevant to the city authorities. The push factors are well documented elsewhere, and we found it not necessary to focus on them. Our focus on studing intra-urban mobility and not rural-urban migration. In this study, we only focus on rural-migrant street children as the primary focus of our research for whom little is known. The choice of our outcome variable is relevant to the field of migration and cities and scholars of migration health studies. 

To the best of our knowledge the policy implications highlighted in the discussion are straight forward and relevant to government and Kampala Capital city Authority in addressing street children and youth’s intra-city mobility and its associated challenges, especially sex work and sexual exploitation. They also suit the local policy and environmental contexts of the study. We highlight recommendations including regulation of sex work in the wake of sex exploitation, which is a big concern in Uganda. Establishing of a longitudinal database for city residents is a strong recommendation and could help address some of the issues which our study was unable to provide - explore the drivers of intraurban mobility to the fullest. 

If mobility is the response to negative push factors, the local government should aim to provide better shelter and reduce the level of street crime. This would then lower the intra-urban mobility of street children and that would be a good thing. However, if intra-urban mobility is the means by which the children can gain a greater income, mobility should be encouraged. The statistical analysis in this paper does not help us to understand what the impact of intra-urban mobility is on the children’s wellbeing.

Response: We would like to state that the impact of intra-urban mobility on children wellbeing is a novel topic but is not the main study question for our research. We set out to investigate the drivers of intra-urban mobility of migrant street children and youth. Perhaps future research could attempt to explore that. We have acknowledged the safety issue of street children and youth in the second last paragraph of the background. Crime was not a major finding of the study, although, we found it important to include in the discussion. 

If the objective of this paper is simply defined as an inquiry into the characteristics of migrant street children and youth, then that can also be a perfectly legitimate research objective (but of lesser importance), provided the analysis is done well and is convincing. However, while this paper has some strong aspects, the analysis is technically flawed.

Response: The importance of this study has been well articulated in the background and the discussion. The analysis was re-done using the negative binomial regression as you suggested. 

The strong aspects of the research reported in this paper are the sampling strategy and the use of mixed methods (analysis of qualitative and quantitative data).

Nonetheless, I concur with reviewer #1 that the statistical analysis is flawed. Let me elaborate. If we define the variable of interest as the number of intra-urban moves that the child or young person makes after arriving in Kampala, this outcomes variable is an integer that takes on the values 0, 1, 2, …. The Poisson regression model can be an appropriate model for quantifying the determinants of this mobility process. At the top of page 8 the authors argue that the Poisson regression assumptions have been satisfied. However, rather than testing for the negative binomial model (and hence overdispersion) as an alternative, they simply note that there is “minimal data dispersion”. But in a Poisson process the mean and the variance are the same, whereas in the Kampala data the variance is almost twice the mean!

Response: We have repeated the analysis using negative binomial regression which best suits our data and taken into consideration other potential predictors while controlling for confounding. However, both Poisson and negative binomial regression models yielded similar findings. 

Even more concerning is, as reviewer #1 notes, that no account is taken of varying exposure across sampled individuals. The Poisson process has the property that if the expected number of events over a period of length T is E, the expected number of events over a period of length 2T is 2E. In other words, the number of months between the interview and the year and month in which the person entered Kampala needs to be taken into account. This information was collected (see Table 1) and should be inserted in the exposure option in the Stata command poisson.

Response: Our data is non-longitudinal and therefore not feasible to account for the time exposure aspects as you suggested. We have re-analysed the data using the three models: Poisson, negative binomial regression and OLS model as suggested by all the reviewers. We present the results from all the two regression models in Table 2. In this regard, the OLS is not our preferred model based since it does not predict the outcome perfectly well compared to the negative binomial regression. However, the negative binomial regression best fits our data and therefore the most appropriate. 

An alternative approach is to calculate the number of moves per month of residence in the city and to run an OLS regression model with this statistic as the dependent variable. For this relatively large sample of 412 observations, this OLS regression is probably not a bad approximation to identifying the statistically significant determinants of intra-urban mobility. The distinction made by the authors between duration of stay > 2 years and duration of stay < 2 years is too coarse.

Response: We have maintained the negative binomial regression as the most appropriate model that fits our data. While the OLS is possible alternative, unfortunately, it does not predict our outcome perfectly. Only one variable (gender) was statistically significant. Thus, we present findings from all three models as proposed by all the three reviewers. Our data is non-longitudinal and does not capture the moves per month of residence over the duration of stay of 15 years. Previous studies in Uganda have used the 2-year cut off for migration status and that is plausible within the context of internal migration in Uganda (see reference).

The numbers of observations that were used in each of the bivariate regressions and in the multivariate regression in Table 2 were actually not stated but in many cases the number was less than 412 due to missing data. I think that this problem could have been overcome by using some form of data imputation, which can be done manually or by means of Stata.

Response: An explanation for the missing values is provided in the data analysis section of the manuscript. 

Another important issue in this context is the assumption of behavioural homogeneity. It is very likely that the Poisson model differs structurally across gender or age (12-17 versus 18-24). This can be tested by running separate regressions for the sub-samples. It is noted on p.8 that further stratification by age is not possible, due to missing data, but imputation may ameliorate this. Ditto for regressions by gender.

Response: Sub-analysis by age and gender did not yield any significant differences and as such, we think this is not worth reporting in the findings. 

Table 2: The number of observations in each of the bivariate regressions and in the multivariate regression should be stated in the Table. Additionally, there are no robustness checks of the multivariate model. For example, log likelihood tests could be used to check the importance of the sexual behaviour-related variables.

Response: Table w has been updated to include multivariate findings from the three analysis models. 

Also, there does not seem to be a full correspondence between the descriptivism in Table 1 and the regressors in Table 2. For example, why is “highest education attained” not in Table 1?

Response: We have included highest education variable in table 2.

Minor points

Abstract: “continuous scale” should be “integer scale”

Abstract: just stating IRR=0.67 for gender does not inform the reader that the mobility is less among girls.

Response: Due to the limitation on word limit, we present the findings with no detailed explanations, which we think is fine for the readers to appreciate the key study findings at a glance. 

Abstract: “causal” should be “casual” By the way, the findings with respect to “Personal safety” and “cost of place of stay” are not shown in the regressions in Table 2.

Response: These are qualitative findings which are derived from data presented in the results section and therefore are not part of the quantitative analysis. These data are important to be included in the abstract. 

p.4, second para.: because the data come from a larger cross-sectional study, there should be a reference to a report or article that describes this larger study, as well as acknowledgement of the funding for this larger study.

Response: The main paper is still in press and therefore cannot be referenced at the moment. 

Table 1: In-school, all respondents: the frequency is 35, not 55.

Response: The correct figures are n=337 for out of school vs n=55 for in-school. This is a valid finding. 

p.16, 4th line from bottom: if intra-urban mobility may not be predicted by duration of stay, the stochastic process is not Poisson!

Response: This has been addressed in the analysis. 

7. PLOS authors have the option to publish the peer review history of their article (what does this mean?). If published, this will include your full peer review and any attached files.

Do you want your identity to be public for this peer review? For information about this choice, including consent withdrawal, please see our Privacy Policy.

Reviewer #1: No

Reviewer #2: No

Reviewer #3: No

---

## [Editor Report · Decision Letter 3]

3 Feb 2021

Demographic and behavioural drivers of intra-urban mobility of migrant street children and youth in Kampala, Uganda

PONE-D-19-29260R3

Dear Dr. Bwambale,

We’re pleased to inform you that your manuscript has been judged scientifically suitable for publication and will be formally accepted for publication once it meets all outstanding technical requirements.

Kind regards,

Andrea Antonio Guido Caragliu

Academic Editor

PLOS ONE

Additional Editor Comments (optional):

Dear Dr. Mulekya, dear Francis:

thank you for resubmitting your work to Plos ONE. I believe your third submission addresses all my remaining concerns and agree with the proposal to only keep the Negative binomial regressions as the core findings of your paper. Please decide on your own whether to leave the two alternative estimates on the paper as robustness checks, perhaps in a technical appendix, or if instead simply leave a reference to these alternative results as a footote.

Thank you all for your fine contribution to the journal,

Kind regards,

Andrea Caragliu
---

## [Editor Report · Acceptance letter]

8 Feb 2021

PONE-D-19-29260R3 

Demographic and behavioural drivers of intra-urban mobility of migrant street children and youth in Kampala, Uganda 

Dear Dr. Bwambale:

I'm pleased to inform you that your manuscript has been deemed suitable for publication in PLOS ONE. Congratulations! Your manuscript is now with our production department. 

Kind regards, 

on behalf of

Professor Andrea Antonio Guido Caragliu 

Academic Editor

PLOS ONE